# A fully integrated optimization framework for designing a complex geometry offshore wind turbine spar-type floating support structure

Mareike Leimeister[1,2], Maurizio Collu[1], and Athanasios Kolios[1]

[1]Naval Architecture, Ocean and Marine Engineering, University of Strathclyde, 100 Montrose Street, Glasgow G4 0LZ, United Kingdom

[2]Division System Technology, Fraunhofer IWES, Institute for Wind Energy Systems, Am Luneort 100, 27572 Bremerhaven, Germany

**Correspondence:** Mareike Leimeister (mareike.leimeister@iwes.fraunhofer.de)

**Abstract.** Spar-type platforms for floating offshore wind turbines are considered suitable for commercial wind farm deployment. To reduce the hurdles of such floating systems to become competitive, a fully integrated optimization framework is applied to design an advanced spar-type floater for a 5 MW wind turbine. Three cylindrical sections with individual diameters and heights, as well as the ballast filling height are the modifiable design variables of the optimization problem. Constraints

5   regarding the geometry, ballast, draft, and system performance are specified, considering alternative materials and allowing for innovative manufacturing solutions. The optimization objective to minimize the floater structural material shall represent the overall goal of cost reduction. Preprocessing system simulations are performed to select a critical design load case, which is used within the iterative optimization algorithm. This itself is executed by means of a fully integrated framework for automated simulation and optimization and utilizes a genetic algorithm. For the applied methodology and conditions it is shown that the

10   required material for such an advanced spar-type platform supporting an offshore wind turbine can be reduced by more than 31% and, at the same time, the performance of the floating system - expressed by the maximum system inclination, maximum tower top acceleration, and mean translational motion - improved in some respect. Thus, the presented design optimization example emphasizes the advantage of following a freer optimization formulation and allowing for novel structural approaches, by which means innovative floater designs, optimized with respect to the global system performance, can be obtained.

**Abbreviations:** AEP, Annual Energy Production; ALPSO, Augmented Lagrangian Particle Swarm Optimization; BC, Base Column; $BC_{low}$, Base Column lower part; $BC_{mid}$, Base Column middle part; $BC_{up}$, Base Column upper part; CapEx, Capital Expenditure; COBYLA, Constrained Optimization BY Linear Approximation; DLC, Design Load Case; DNV GL, Det Norske Veritas and Germanischer Lloyd; Dymola, Dynamic Modeling Laboratory; IEC, International Electrotechnical Commission; IWES, Institute for Wind Energy Systems; LCoE, Levelized Cost of Energy; MoWiT, Modelica library for Wind Turbines; NREL, National Renewable Energy Laboratory; NSGAII, Non-dominated Sorting Genetic Algorithm II; NSGAIII, Non-dominated Sorting Genetic Algorithm III; OC3, Offshore Code Comparison Collaboration; OC4, Offshore Code Comparison Collaboration Continuation; OpEx, Operational Expenditure; Rkfix4, Runge-Kutta fixed-step and 4th order method; SPEA2, Strength Pareto Evolutionary Algorithm 2; SWL, Still Water Level; TI, Turbulence Intensity; TP, Tapered Part; UC, Upper Column

# 1 Introduction

With floating support structures for offshore wind turbines, more offshore wind resources can be captured and used for power generation, as around 60% to 80% of the ocean areas cannot be exploited with bottom-fixed structures, which are limited to water depths of up to around 50 m (European Wind Energy Association, 2013). The floating offshore wind technology is no longer in its infancy. Over the last decade, the technology readiness level of floating offshore wind turbine systems has significantly increased so that "floating offshore wind is coming of age", as WindEurope states in its floating offshore wind vision statement (WindEurope, 2017, p.4). The large number of research studies, research projects, scaled model tests, prototype developments, and full scale model test phases paved the way towards this current status. More than 40 floating foundation concepts exist and are under development (Quest Floating Wind Energy, 2020; Future Power Technology, 2019; James and Ros, 2015; Mast et al., 2015). A few selected milestones are (Löfken, 2019)

- the Hywind spar-buoy floating system, with a 2.3 MW demonstrator deployed in 2009, the subsequent Hywind Scotland pilot park of five 6 MW turbines operating since 2017, and another wind farm Hywind Tampen with eleven 8 MW turbines planned for 2022;

- the WindFloat semi-submersible floating system by Principle Power, with three 2 MW demonstrators since 2011 and the floating wind farm WindFloat Atlantic with three 8.4 MW turbines completed in 2020[1];

- the Damping Pool® (Floatgen) barge floating system by Ideol, with a 2 MW and a 3 MW demonstrator since 2018 and further large projects with for example 6.2 MW wind turbines planned for the future; and

- the TetraSpar spar, semi-submersible, or tension leg platform floating system by Stiesdal Offshore Technologies, with a demonstrator supporting a 3.6 MW wind turbine planned for spring 2021.

Despite the great amount of floating offshore wind projects, most of them are under development and currently the Hywind Scotland pilot park and WindFloat Atlantic[1] are the only operational floating wind farms (Future Power Technology, 2019), apart from the first prototype floating wind farm within the Fukushima Floating Offshore Wind Farm Demonstration Project FORWARD, in which three different floating wind turbines connected to the same floating substation are tested for a limited operating life (James and Ros, 2015; Main(e) International Consulting LLC, 2013). More are planned as already mentioned above; however, for further speed-up of the market uptake of floating offshore wind farms, significant cost-reductions are still required. From the survey-based study by Leimeister et al. (2018) the conclusion is drawn that the spar-buoy concept is the most mature and has the highest technology readiness level. However, in order to enhance its suitability for multi-MW wind farm deployment, this technology has to be further advanced: The common spar-buoy floater is already very convenient for volume production and certification due to its simple geometry; however, to facilitate an accelerated and global market uptake, especially the large floater draft has to be reduced so that in the end the levelized cost of energy (LCoE) is reduced and the handling is simplified.

---

[1]https://www.edp.com/en/innovation/windfloat (Accessed: 13 March 2021)

To overcome the challenges that the highly promising spar-buoy floating platform type still faces, a few researchers have already worked on concepts for advanced spar-type floating offshore wind turbine support structures, which have a reduced draft but still provide sufficient stability and do not necessarily consist of just one long cylindrical element (Hegseth and Bachynski, 2019; Wright et al., 2019; Yoshimoto and Kamizawa, 2019; Zhu et al., 2019; Hirai et al., 2018; Yoshimoto et al., 2018; Yamanaka et al., 2017; Matsuoka and Yoshimoto, 2015; Lee, 2005). However, different approaches for designing the floating platform are followed and it does not seem that a fully integrated optimization approach is adopted. Other design development studies (Chen et al., 2017; Perry et al., 2007; Bangs et al., 2002) are inspired by the oil and gas industry and deal with so-called truss spar platforms, in which a truss section connects a bottom tank with the floating platform and heave plates can be included. However, only Perry et al. (2007) apply a genetic algorithm based optimization for developing a cost-efficient preliminary floating support structure design.

Some other researchers focus on the optimization of the dynamic response of the floating offshore wind turbine system by rather adding and optimizing additional components instead of the spar-type structure itself. Hence, Ding et al. (2017b, a) use helical strakes - again inspired by the oil and gas industry - and a heave plate, while He et al. (2019) optimize a tuned mass damper by utilizing an artificial fish swarm algorithm. Pham and Shin (2019) add a moonpool, which is optimized together with the commonly shaped spar-type platform, following a three-step and, hence, no integrated optimization approach.

The majority of design optimization approaches, however, is based on the common spar-type floater shape and utilizes gradient-based methods (Dou et al., 2020; Hegseth et al., 2020; Berthelsen et al., 2012; Fylling and Berthelsen, 2011) or genetic algorithms (Karimi et al., 2017; Choi et al., 2014). Some applications are purely dealing with the support structure - focusing on basic hydrodynamic analyses, maximum system stability, and minimum material cost (Choi et al., 2014), reduced draft, weight, and cost with at the same time increased power output (Lee et al., 2015), or optimized floater cost and power generation (Gao and Sweetman, 2018) - while other design optimization approaches are highly complex and account for optimizing several components of the floating wind turbine system, such as the tower, mooring system, power cable, and/or blade-pitch controller in addition to the floating platform, and focus on extreme loads, structural strength, fatigue life, or power quality in addition to costs and global system responses (Dou et al., 2020; Hegseth et al., 2020; Sandner et al., 2014; Fylling and Berthelsen, 2011) or distinguish also between different floater types (Karimi et al., 2017; Sclavounos et al., 2008).

Even if a reduced draft is often aimed and obtained (Hegseth et al., 2020; Gao and Sweetman, 2018; Lee et al., 2015; Sandner et al., 2014) and sometimes the spar-buoy floater is subdivided into several cylindrical sections (Hegseth et al., 2020; Berthelsen et al., 2012; Fylling and Berthelsen, 2011) or a broad range of allowable values is considered for the design variables (Karimi et al., 2017; Sclavounos et al., 2008), always common spar-type platform designs are considered, meaning a structure consisting of welded sections, for which reason even Hegseth et al. (2020) limit the maximum allowable taper angle. Thus, the aim of this paper is to demonstrate that, through a more comprehensive fully integrated design optimization approach and by allowing design variables out of a wider range of values, more potential solutions for an advanced spar-type floater design can be captured. Apart from reducing the floater draft, the main objective is cost reduction - expressed in terms of the material used - while global system performance criteria have to be fulfilled. All these requirements regarding design variables and optimization criteria are - together with specific environmental conditions and the fully-coupled aero-hydro-servo-elastic dy-

namic characteristics of a floating offshore wind turbine system - incorporated into a fully integrated optimization framework. By means of this, an advanced spar-type floating offshore wind turbine support structure design is aimed to be obtained. The focus of the optimization procedure lies on hydrodynamic and system-level analyses and not that stringent limitations on the structure and dimensions are required. This way and by considering novel structural realization approaches for the resulting optimized geometries, as well as different ballast materials, new alternatives of potential and innovative floater design solutions are opened up.

In order to figure out in detail the required characteristics of such a floating platform, first, advanced spar-type floating wind turbine support structures are elaborated in detail in Sect. 2 and a reference floating system with corresponding assessment criteria is specified. Based on this, the optimization problem - consisting of design variables, objective function, and constraints - is defined in Sect. 3. Subsequently, the automated design optimization of the advanced spar-type floating wind turbine system is performed in Sect. 4, including some preprocessing automated design load case (DLC) simulations, as well as the characterization of the automated optimization framework and the iterative optimization approach. The results of the optimization simulations are presented in Sect. 5 and further discussed in Sect. 6. Finally, some conclusions are drawn in Sect. 7.

## 2    Advanced spar-type floating wind turbine support structures

According to the survey conducted by Leimeister et al. (2018), industry professionals and scientific experts judge the advanced spar-type floating platform - compared to the common spar-buoy floaters, semi-submersibles, tension leg platforms, barges, or any hybrid, multi-turbine, or mixed-energy floating system - to be the most suitable wind turbine support structure for deployment in floating offshore wind farms. This selection of the spar-type floating platform is made in consequence of its suitability for serial production, possibility of receiving certification, low LCoE, and little demands on the mooring system.

### 2.1    Characteristics of advanced spar-type floaters

The common spar-buoy floating platform consists of a long relatively slender cylinder which is filled at the bottom end with ballast. The resulting deep center of gravity provides stability against overturning. However, this floating system exhibits some weaknesses: Due to its deep draft it cannot be deployed in shallow or intermediate waters up to around 100 m water depth (James and Ros, 2015), nor can the entire floating wind turbine system be fully assembled in upright position onshore or at harbor sites. The latter fact adds to the already expensive floater, as it makes the overall handling of this long and heavy structure, its assembly, transport, and installation costly. Thus, by

– reducing the draft,

– applying a delta or so called crowfoot connection of the mooring lines to the spar-buoy structure, and/or

– adding damping fins,

the advanced spar-type floating system can benefit from

- a wider range of possible installation sites,

- simplified handling (both construction, assembly, transport, and installation),

- reduced system, as well as construction and transportation costs, as well as

- improved system motion performance. (Leimeister et al., 2018)

In particular, these characteristics of advanced spar-type floating platforms are realized in a few - both research and real - concepts. The advanced spar-type floater by the Massachusetts Institute of Technology (Lee, 2005) has a relatively shallow draft and gets stability support from a two-layered taut-leg mooring system (Butterfield et al., 2007). Both Hirai et al. (2018) and Yamanaka et al. (2017) use a three-segmented advanced geometry spar, where a larger diameter column makes up the middle part to allow for shortening the overall length of the spar and reducing the system cost. In contrast, Zhu et al. (2019) utilize the three elements just in an opposite way, focusing on increased restoring and improved motion performance: The spar element makes up the middle part and interconnects two columns, one with just a slightly larger diameter at the bottom end and another one with a large diameter at the upper end.

Within the Fukushima Floating Offshore Wind Farm Demonstration Project FORWARD an advanced spar-type support structure, developed by Japan Marine United, is utilized for a floating substation (Fukushima Kizuna) and a 5 MW wind turbine (Fukushima Hamakaze) (Yoshimoto and Kamizawa, 2019; James and Ros, 2015; Main(e) International Consulting LLC, 2013). The advanced spar for the floating substation consists of three columns - or so called hulls - placed at the bottom, in the middle, and at the upper end (intersecting the water line) of the spar. This way, the floating system is suitable already at around 110 m water depth, the motion performance is improved, and the cost for installation is reduced (Wright et al., 2019; Yoshimoto et al., 2018; Matsuoka and Yoshimoto, 2015). The Fukushima Hamakaze was initially using a similarly structured advanced spar, equipped with damping fins for stabilization in sway and heave direction (James and Ros, 2015; Main(e) International Consulting LLC, 2013); however, after some investigations and studies by Matsuoka and Yoshimoto (2015), finally, the advanced spar-type platform for the 5 MW wind turbine consists of just two large columns/hulls at the bottom and top end of the spar and, thus, is optimized with respect to the system restoring and motion performance, as well as the construction cost (Yoshimoto and Kamizawa, 2019). Despite these optimizations, the installation of the floating platform - in particular the ballasting operations - turned out to be complex, as the floater has leaned to an angle of $45°$ when it was brought from the construction draft to a deeper draft. This issue, however, could be resolved within less than a week (JWPA, 2017; Foster, 2016).

Within this study, however, the definition of an advanced spar-type floater is further extended and goes beyond the main objectives to reduce the draft of the floater and the cost of the overall system. Thus, additionally, alternative materials are investigated, which are from an economic point of view comparative to currently used materials, however, positively influence the final floater design due to their different material properties and characteristics. Furthermore, the term advanced spar-type floater - used in this study - not only addresses the floating structure itself, but also includes the consideration of novel structural approaches which might be more promising than the common approach of welding cylindrical and tapered sections

together and allow a widening of the design space for such innovative and advanced floater designs. The specific steps taken for addressing the definition of an advanced spar-type floater in a broader sense are described in detail in Sect. 2.4.

## 2.2 Reference floating offshore wind turbine system

As starting point of the design optimization towards an advanced spar-type floating platform for an offshore wind turbine, a
150 traditional spar-buoy floating wind turbine system, taken from phase IV of the OC3 (Offshore Code Comparison Collaboration) project (Jonkman, 2010), is used. This is further modified, as explained in Sect. 2.3, to allow the development of an advanced spar-type structure.

The OC3 phase IV floating offshore wind turbine system consists of

- the NREL (National Renewable Energy Laboratory) 5 MW reference wind turbine (Jonkman et al., 2009) with a rotor
diameter of 126.0 m, a hub height of 90.0 m, an overall mass of the rotor-nacelle assembly of 350,000 kg, and an operating range between 3.0 m/s and 25.0 m/s with a rated wind speed at 11.4 m/s;

- an offshore adapted tower (Jonkman, 2010), which ranges from 10.0 m elevation above still water level (SWL) up to 87.6 m, is tapered from a diameter of 6.5 m and wall thickness of 0.027 m at the base to a diameter of 3.87 m and wall thickness of 0.019 m at the top, and weighs 249,718 kg;

- the spar-buoy floater (Jonkman, 2010) with a top diameter of 6.5 m at 10.0 m elevation above SWL, which increases between 4.0 m below SWL and 12.0 m below SWL up to 9.4 m - corresponding to the diameter at the base at 120.0 m below SWL - and an overall mass - including ballast - of 7,466,330 kg; and

- three evenly spaced catenary mooring lines of 902.2 m length each and 0.09 m diameter, which are attached to the spar-buoy at 70.0 m below SWL and anchored to the seabed at 320.0 m water depth in a radius of 853.87 m from the floater
centerline (Jonkman, 2010).

## 2.3 Modifications for defining an advanced spar-type floater

An aero-hydro-servo-elastic coupled model of dynamics of this original OC3 phase IV spar-buoy floating wind turbine system is developed and verified by Leimeister et al. (2020a), using MoWiT (Modelica library for Wind Turbines) (Leimeister and Thomas, 2017; Thomas et al., 2014; Strobel et al., 2011). The modeling approach in MoWiT utilizes the object-oriented,
equation-based, and component-based modeling language Modelica[2] and, therefore, follows a hierarchical structure with interconnected main components and subcomponents to represent the complex wind turbine system and fully-coupled system dynamics. This multibody approach provides high flexibility to model various wind turbine system types, environmental conditions, and simulation settings by simply modifying single model components. MoWiT, developed at Fraunhofer IWES (Institute for Wind Energy Systems), is continuously enhanced and more features and interfaces are added, which opens up a broad ap-

---

[2]https://www.modelica.org/ (Accessed: 22 January 2020)

plication range, including the integration into a framework for automated simulation and optimization, as utilized in this study. (Leimeister et al., 2021)

This MoWiT model of the OC3 phase IV spar-buoy floating wind turbine system is used as basis and modified so that a design of an advanced spar-type floater can be obtained through automated optimization. As this work focuses on the design of the floating platform and not on the mooring system, a shorter, less heavy, and, hence, cheaper advanced spar-type floater design

shall be obtained by changing the floater geometry. Different characteristic shapes of advanced spar-type floating platforms are pointed out in Sect. 2.1. In this study, a similar concept as presented by Zhu et al. (2019) and realized in the Fukushima Hamakaze floating wind turbine system (Matsuoka and Yoshimoto, 2015; Yoshimoto and Kamizawa, 2019) is applied: The long cylindrical element below the tapered part is divided into three partitions:

1. the base column upper part $BC_{up}$, which shall serve for gaining buoyancy;

2. the base column middle part $BC_{mid}$, which mainly provides the separation of parts 1 and 3 to deepen the position of part 3; and

3. the base column lower part $BC_{low}$, which can be filled with ballast and this way shall shift the center of gravity downwards.

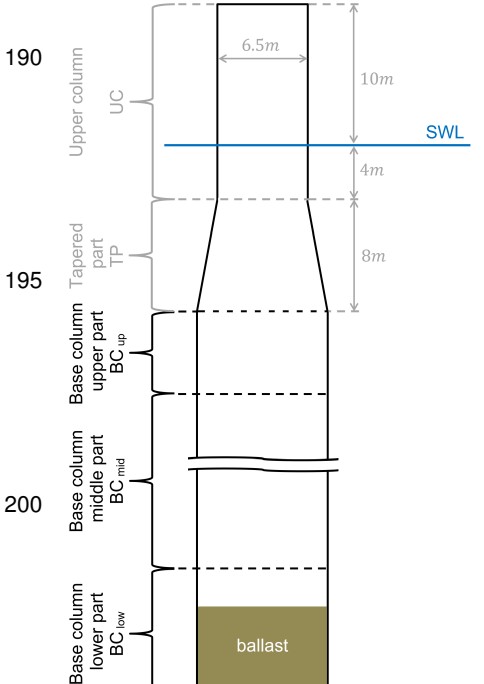

**Figure 1.** Geometrical definitions of the advanced spar-type floating platform.

This partitioning is schematically represented in Fig. 1, showing the unchanged geometric parameters and dimensions for the upper column (UC) and tapered part (TP) in a light shade (gray) and indicating the three sections of the base column (BC) together with the ballast filling in the base column lower part $BC_{low}$. In order to still represent the original OC3 phase IV floating spar-buoy with the modified MoWiT model, initially the diameters of all three BC parts are set equal to the original spar diameter of 9.4 m. Furthermore, as a ballast filling is just allowed in $BC_{low}$, the heights of $BC_{up}$ and $BC_{mid}$ are set equal to machine epsilon, which corresponds to a value of $10^{-15}$ in Modelica, while $BC_{low}$ holds the full original length of 108.0 m. Regarding the hydrodynamic coefficients for the three cylindrical partitions, the same as for the original OC3 phase IV spar-buoy (Jonkman, 2010) are applied.

Apart from these modifications, which are directly related to an advanced spar-type floater design, also the material density of the support structure and the wall thickness of the cylindrical spar-buoy elements are changed. As the material density of the OC3 phase IV spar-buoy is not explicitly defined in the definition document (Jonkman, 2010), a value of 10,000 kg/m$^3$ is derived in the model verification study (Leimeister et al., 2020a). However, to better match the common steel properties of offshore structures, a material density of 7,850 kg/m$^3$ is used for the design of the

advanced spar-type floating platform. Furthermore, the wall thickness of the spar structure[3] is changed from the fixed value of 0.0314 m, which is derived by Leimeister et al. (2020a), to a wall thickness that is adaptable to the specific advanced spar-type floater design. In order to obtain an appropriate wall thickness for a corresponding floater design, a fixed ratio of the support structure structural mass to the displaced mass of water is deployed. This ratio is for a spar-type floating platform 0.13 - according to representative values from research designs and academic studies and excluding designs, such as the Hywind demonstrator, which are for safety reasons heavily oversized (Bachynski, 2018). Hence, the equivalent structural mass of the advanced spar-type floater (meaning the mass of the advanced spar-type steel structure, excluding the tower and wind turbine, and also excluding the ballast mass) with certain outer dimensions (diameters $D_i$ and heights $H_i$) and corresponding displaced volume can be determined following Eq. (1).

$$\frac{\text{spar structural mass}}{\text{buoyancy mass}} = 0.13 \tag{1}$$

With the resulting structural mass of the advanced spar-type floater of 1.070E+06 kg, which is a bit lower than the original structural mass of 1.150E+06 kg (Leimeister et al., 2020b), the corresponding appropriate wall thickness, which is kept the same and constant for all parts of the specific advanced geometry spar design, is computed by means of Eq. (2). This equation is derived from the expression for the mass of the advanced geometry spar steel structure with a material density of 7,850 kg/m$^3$ as explained above. In Eq. (2), $H_i$ and $D_i$ are the heights and diameters of each element, meaning UC, TP, BC$_{\text{up}}$, BC$_{\text{mid}}$, and BC$_{\text{low}}$. The diameter of the tapered part $D_{\text{TP}}$, however, is determined according to Eq. (3) as mean of the diameters of UC and BC$_{\text{up}}$.

$$\text{wall thickness} = \frac{\sum_i (H_i D_i) - \sqrt{\left[\sum_i (H_i D_i)\right]^2 - \frac{4}{\pi} \frac{\text{spar structural mass}}{\text{material density}} \sum_i H_i}}{2 \sum_i H_i} \tag{2}$$

$$D_{\text{TP}} = \frac{D_{\text{UC}} + D_{\text{BC,up}}}{2} \tag{3}$$

This way, a wall thickness of 0.0372 m is obtained for the original OC3 phase IV spar-buoy with reduced material density (7,850 kg/m$^3$) and adopted structural mass to displaced mass ratio of 0.13. This wall thickness value lies within the acceptable range, based on available data for the semi-submersible floating platform from phase II of OC4 (Offshore Code Comparison Collaboration Continuation).

As the advanced spar-type floater design (optimization) study does not focus on the mooring system, as mentioned above, and due to the fact that the mooring system itself could be covered in a separate optimization task, any change in the restoring system characteristics due to shifted fairlead positions is prevented by utilizing constant (the original) resulting mooring system properties. This means that - independent of possible attachment points to the reshaped floating platform - the resulting stiffness

---

[3]Referring here purely to the circumferential walls of the hollow cylindrical or conical elements, as for base and lid a fixed marginal cap thickness of 0.001 m is applied, according to the implemented model in the verification study (Leimeister et al., 2020a).

of each mooring line is taken from the system motion, assuming the original fairlead positions as defined in Sect. 2.2. A realistic mooring system design for the finally obtained optimized floating platform, which represents the considered resulting mooring system properties, can then afterwards be obtained through a subsequent optimization. This might even happen manually - depending on the degree of complexity - as it is applied in studies for designing equivalent mooring systems (Molins et al., 2015; Udoh, 2014). However, having not included the mooring system as design variable within the optimization of the floating spar-type platform, further system performance improvements due to modified mooring system parameters or fairlead positions - in addition to an optimized support structure design - are limited. This, however, leaves open the possibility of subsequent fine tuning of the design solution obtained through optimization based on hydrodynamic and system-level analyses. By addressing the mooring system in a successive but separate optimization algorithm, the dynamic response of the floating offshore wind turbine system, as well as the mooring line tension itself, can be significantly improved by considering an advanced and more complex optimization problem, in which - apart from various line diameters and lengths - different mooring line arrangements and distribution forms can be utilized, the optimum number of lines within the mooring system and best fairlead position elaborated, different mooring types used or even mixed within segmented lines, and also clump weights incorporated (Tafazzoli et al., 2020; Barbanti et al., 2019; Men et al., 2019; Chen et al., 2017).

## 2.4 Assessment criteria for designing an optimized advanced spar-type floater

The focus in this study lies on obtaining an advanced spar-type floating platform, which is characterized through a limited draft and reduced structural cost, but still shows good hydrodynamic performance. Any detailed structural integrity checks are not addressed in this work, but can be added for a more extensive optimization approach. However, by focusing only on hydrodynamics and global system performance without defining any restrictions regarding structural aspects, floater designs, which would have been discarded when performing structural integrity checks and as they would be unfeasible to be realized with conventional structural approaches, can still be captured as potential solutions when considering different structural realization approaches.

The only structural related focus, considered in this approach, is the minimization of the structural cost. This is represented through the steel volume of the floater, which is finally specified as objective of the optimization problem, as formally declared in Sect. 3.2.

In order to achieve the shortened length of the advanced spar-type floater, the allowable draft of the system is limited to the original draft of the OC3 phase IV floating wind turbine system as maximum value, as well as to a recommended minimum value of 15.0 m (Ng and Ran, 2016). The resulting allowable total height of the BC has to be distributed to the three partitions; however, no restrictions prevail and also the option of utilizing not all three BC parts is possible. Thus, the minimum allowable value for the height of each of the BC parts is machine epsilon ($10^{-15}$ m) - as a zero value is unfeasible from a modeling point of view. For the ballast height, it additionally has to be guaranteed that it does not exceed the actual $BC_{low}$ height. The resulting allowable value ranges based on the draft limits are summarized in Table 1.

The applied concept of a three-segmented advanced spar-type floater with elements for buoyancy, distance, and ballast shall not only allow different heights but also different diameters of these elements. Thus, the allowable value range for the

**Table 1.** Allowable value ranges addressing the draft limits.

| | Allowable draft | Resulting total height of BC | $BC_{up}$ height | $BC_{mid}$ height | $BC_{low}$ height | Ballast height |
|---|---|---|---|---|---|---|
| **Min** | 15.0 m | 3.0 m | $10^{-15}$ m | $10^{-15}$ m | $10^{-15}$ m | $10^{-15}$ m |
| **Max** | 120.0 m | 108.0 m | 108.0 m | 108.0 m | 108.0 m | 108.0 m |

diameter of each of the BC parts is set from machine epsilon - due to the same modeling feasibility reason - to 120.0 m. The maximum diameter is chosen deliberately large - corresponding to the total maximum draft of the floating system - to ensure that the border of feasible solutions is well captured. From a manufacturing point of view, cylindrical offshore structures with diameters of more than 10.0 m are realistic: Various sources[4,5] state a value of 11.0 m, the reference semi-submersible floating platform from phase II of OC4 has an upper column diameter of 12.0 m (Robertson et al., 2014), and the diameter

of the spar-buoy utilized in the Hywind Scotland floating wind farm[6,7] is even up to 14.5 m large. However, looking at other floating platform solutions, such as the Damping Pool® floater by Ideol[8] with outer dimensions of 36 m x 36 m and a resulting diagonal length of almost 51 m or again the OC4 phase II semi-submersible platform (Robertson et al., 2014) with an overall outer dimension of almost 82 m in diameter, shows that floating structures with a large overall outer diameter can be obtained without being restricted to the manufacturing feasibility limits for pure cylinders. Thus, from a hydrodynamic point of view,

a cylindrical offshore structure with very large diameter can be realized as well through several smaller diameter cylinders being clustered together in a circle, representing similar hydrodynamic behavior and characteristics. Finally, attention has to be drawn on the minimum possible diameter of the BC parts, which always has to be at least as large as twice the actual wall thickness corresponding to the specific advanced spar-type floater design.

Having modified the diameters and heights of the three BC parts, as well as the ballast filling height, and having adjusted

the wall thickness according to the structural mass to displaced mass ratio, as defined in Sect. 2.3, the ballast density has to be adjusted to match the original floating equilibrium between buoyancy force, system weight, and downward mooring force, so that the original hub height is maintained. In order to exclude unfeasible system solutions, in which material would have to be removed from the system (realized for example by reducing the material density) to meet this equilibrium condition, it has to be ensured that the actual resulting ballast density of the specific advanced spar-type floater design carries a positive

value. However, in order to account for truly realistic ballast densities, also the uppermost allowable value of the ballast density has to be constrained. Leimeister et al. (2020b) have explored, within a first-stage design optimization application example, densities for common and cheap materials to be used as ballast for a floating spar-buoy. The densest material included is

[4]https://sif-group.com/en/wind/foundations (Accessed: 13 August 2019)

[5]https://www.windkraft-journal.de/2019/06/14/steelwind-nordenham-ist-von-wpd-die-gruendungsstrukturen-fuer-den-offshore-wind-park-yunlin-in-taiwan-zu-fertigen/136551 (Accessed: 13 August 2019)

[6]https://www.equinor.com/content/dam/statoil/documents/newsroom-additional-documents/news-attachments/brochure-hywind-a4.pdf  (Accessed: 13 June 2019)

[7]https://www.equinor.com/en/news/worlds-first-floating-wind-farm-started-production.html (Accessed: 13 June 2019)

[8]https://floatgen.eu/ (Accessed: 13 August 2019)

sandstone (or other rocks) with a density of about 2.6E+03 kg/m$^3$. Apart from sand, sand mixed with water, concrete, or rocks, MagnaDense (heavyweight concrete) is as well used in industry as high density material[9,10,11]. With MagnaDense densities of up to 5.0E+03 kg/m$^3$ can be obtained[12] (LKAB Minerals, 2019). Even if minimization of the structure material volume is defined as objective function - as stated at the beginning of this section - in order to represent the structural cost, the cost of the two potential densest ballast materials is elaborated to avoid significant larger ballast costs when utilizing MagnaDense instead of the common cheap materials pointed out by Leimeister et al. (2020b). However, when comparing the material prices for sandstone[13] (for the ballast density limit of 2.6E+03 kg/m$^3$) and MagnaDense[9,14] (for the ballast density limit of 5.0E+03 kg/m$^3$), it turns out that both ballast materials have a similar cost of around 150 $ per ton, which is less than 20% of the material cost for structural (raw) steel of about 700 $ per tonne[15] (Grogan, 2018; Butcher, 2018). Thus, the ballast density is constrained to a maximum of 5.0E+03 kg/m$^3$.

Apart from these more geometry and material related assessment criteria, there are three performance related criteria, which the advanced spar-type floating offshore wind turbine system has to fulfill. For the global system performance of a floating offshore wind turbine, maximum allowable values are prescribed for

1. the total inclination angle of the system to the vertical:

   For system rotational stability reasons a maximum total inclination angle of 10.0° is allowed in operational conditions (Leimeister et al., 2020b; Katsouris and Marina, 2016; Kolios et al., 2015; Huijs et al., 2013);

2. the total horizontal acceleration at the tower top:

   Due to sensitive components in the nacelle and to prevent any issues with the lubrication, the nacelle acceleration - corresponding to the acceleration at the tower top - is limited, depending on the specific wind turbine, to a maximum of 0.2 to 0.3 times the gravitational acceleration constant (Nejad et al., 2017; Huijs et al., 2013; Suzuki et al., 2011); herein the lower value of 1.962 m/s$^2$ is used following a conservative approach (Leimeister et al., 2020b);

3. the mean translational motion of the floating system:

   Based on experience, the static translational displacement of a (non TLP-type) floating offshore wind turbine system, corresponding to the mean of the translational motion, is restricted to 0.2 times the water depth (320.0 m in the case of the OC3 phase IV spar-buoy floating system), and hence to 64.0 m in this application (Leimeister et al., 2020b).

---

[9]Floating offshore wind project manager at a leading company in offshore industry, personal communication, 6 February 2020.

[10]https://www.lkabminerals.com/en/industry-uses/offshore-energy/offshore-wind-structures/ (Accessed: 7 June 2020)

[11]https://www.lkabminerals.com/de/floating-offshore-wind-2018/ (Accessed: 7 June 2020)

[12]https://www.lkabminerals.com/wp-content/uploads/2019/02/MagnaDense-SDS-12-06INT-19-03.pdf (Accessed: 5 February 2020)

[13]https://www.alibaba.com/showroom/sandstone-price-per-ton.html (Accessed: 5 February 2020)

[14]https://german.alibaba.com/product-detail/magnadense-heavy-concrete--172429386.html (Accessed: 5 February 2020)

[15]https://spendonhome.com/structural-steel-fabrication-cost/ (Accessed: 5 February 2020)

## 3 Definition of the optimization problem

For obtaining an optimized advanced spar-type floater design, following the assessment criteria - as outlined in Sect. 2.4 - and using the modified floating wind turbine system model - as described in Sect. 2.3 - as basis, an iterative optimization approach (explained in more detail in Sect. 4.3) is carried out in this study. This optimization approach requires the definition of the optimization problem - comprising the modifiable design variables $x_i$, the objective functions $f_i$ to be minimized, as well as the equality ($h_i$) and inequality ($g_i$) constraints to be fulfilled - as given in formal expressions in the following:

find $\qquad X = \{x_1, ..., x_k\}$

to minimize $\quad f_i(X, system(X)) \qquad , i = 1, ..., l$

subject to $\quad h_i(X, system(X)) = 0 \quad , i = 1, ..., m$

subject to $\quad g_i(X, system(X)) \leq 0 \quad , i = 1, ..., n$

The functions are either directly dependent on the design variables or also on the resulting fully-coupled floating offshore wind turbine system, denoted with $system(X)$.

### 3.1 Design variables of the advanced spar-type floating wind turbine system

Based on the derivation of the modified spar-buoy floater model for enabling the design of an advanced spar-type floating platform (Sect. 2.3), the design variables vector $X = \{x_1, x_2, ..., x_6, x_7\}$ with seven ($k = 7$) elements is defined as presented in Table 2.

**Table 2.** Definition of the seven design variables.

| Design variable | Formal expression | Description | Allowable value range | Corresponding constraints |
|:---:|:---:|:---|:---:|:---:|
| $x_1$ | $D_{\mathrm{BC_{up}}}$ | Diameter of $\mathrm{BC_{up}}$ | [$10^{-15}$ m, 120.0 m] | $g_1, g_2$ |
| $x_2$ | $D_{\mathrm{BC_{mid}}}$ | Diameter of $\mathrm{BC_{mid}}$ | [$10^{-15}$ m, 120.0 m] | $g_3, g_4$ |
| $x_3$ | $D_{\mathrm{BC_{low}}}$ | Diameter of $\mathrm{BC_{low}}$ | [$10^{-15}$ m, 120.0 m] | $g_5, g_6$ |
| $x_4$ | $H_{\mathrm{BC_{up}}}$ | Height of $\mathrm{BC_{up}}$ | [$10^{-15}$ m, 108.0 m] | $g_7, g_8$ |
| $x_5$ | $H_{\mathrm{BC_{mid}}}$ | Height of $\mathrm{BC_{mid}}$ | [$10^{-15}$ m, 108.0 m] | $g_9, g_{10}$ |
| $x_6$ | $H_{\mathrm{BC_{low}}}$ | Height of $\mathrm{BC_{low}}$ | [$10^{-15}$ m, 108.0 m] | $g_{11}, g_{12}$ |
| $x_7$ | $H_{\mathrm{ballast}}$ | Height of the ballast | [$10^{-15}$ m, 108.0 m] | $g_{13}, g_{14}$ |

### 3.2 Objective function for the advanced spar-type floating wind turbine system

As stated in Sect. 2.4, just one objective function ($l = 1$) is specified, which corresponds to the structure material volume of the advanced spar-type floating platform. This objective function ($f_1$) is to be minimized, as defined at the beginning of Sect. 3.

### 3.3 Constraints for the advanced spar-type floating wind turbine system

Section 2.4 covers already the assessment criteria for designing an optimized advanced spar-type floating platform. These make up - apart from the objective function - 25 constraints, which are all specified as inequality constraints - hence, $m = 0$ (for the equality constraints $h_i$) and $n = 25$ (for the inequality constraints $g_i$). These shall all take on values less or equal to zero, as expressed at the beginning of Sect. 3. The definitions of the inequality constraints are listed in Table 3.

**Table 3.** Definition of the 25 inequality constraints.

| Inequality constraint | Formal expression | Description |
|---|---|---|
| $g_1$ | $10^{-15}$ m $- x_1$ | Allowable value range of $x_1$ |
| $g_2$ | $x_1 - 120.0$ m | Allowable value range of $x_1$ |
| $g_3$ | $10^{-15}$ m $- x_2$ | Allowable value range of $x_2$ |
| $g_4$ | $x_2 - 120.0$ m | Allowable value range of $x_2$ |
| $g_5$ | $10^{-15}$ m $- x_3$ | Allowable value range of $x_3$ |
| $g_6$ | $x_3 - 120.0$ m | Allowable value range of $x_3$ |
| $g_7$ | $10^{-15}$ m $- x_4$ | Allowable value range of $x_4$ |
| $g_8$ | $x_4 - 108.0$ m | Allowable value range of $x_4$ |
| $g_9$ | $10^{-15}$ m $- x_5$ | Allowable value range of $x_5$ |
| $g_{10}$ | $x_5 - 108.0$ m | Allowable value range of $x_5$ |
| $g_{11}$ | $10^{-15}$ m $- x_6$ | Allowable value range of $x_6$ |
| $g_{12}$ | $x_6 - 108.0$ m | Allowable value range of $x_6$ |
| $g_{13}$ | $10^{-15}$ m $- x_7$ | Allowable value range of $x_7$ |
| $g_{14}$ | $x_7 - 108.0$ m | Allowable value range of $x_7$ |
| $g_{15}$ | $\max(\text{total inclination angle}) - 10.0°$ | Maximum total inclination angle |
| $g_{16}$ | $\max(\text{horizontal nacelle acceleration}) - 1.962 \text{ m/s}^2$ | Maximum horizontal nacelle acceleration |
| $g_{17}$ | $\text{mean}(\text{translational motion}) - 64.0$ m | Mean translational motion |
| $g_{18}$ | $3.0 \text{ m} - (x_4 + x_5 + x_6)$ | Minimum draft |
| $g_{19}$ | $x_4 + x_5 + x_6 - 108.0$ m | Maximum draft |
| $g_{20}$ | $x_7 - x_6$ | Ballast filling height within $\text{BC}_{\text{low}}$ |
| $g_{21}$ | $-\text{ballast density}$ | Allowable value range of the ballast density |
| $g_{22}$ | $\text{ballast density} - 5.0\text{E}+03 \text{ kg/m}^3$ | Allowable value range of the ballast density |
| $g_{23}$ | $0.5 \cdot 10^{-15} \text{ m} + \text{wall thickness} - 0.5x_1$ | Wall thickness and diameter of $\text{BC}_{\text{up}}$ |
| $g_{24}$ | $0.5 \cdot 10^{-15} \text{ m} + \text{wall thickness} - 0.5x_2$ | Wall thickness and diameter of $\text{BC}_{\text{mid}}$ |
| $g_{25}$ | $0.5 \cdot 10^{-15} \text{ m} + \text{wall thickness} - 0.5x_3$ | Wall thickness and diameter of $\text{BC}_{\text{low}}$ |

## 4 Automated design optimization of the advanced spar-type floating wind turbine system

The final automated design optimization of the reference advanced spar-type floating wind turbine system described in Sect. 2.3 consists of

1. preprocessing automated system simulations for identifying the simulation conditions to be considered within the optimization (Sect. 4.1), as well as

2. the actual iterative optimization approach for obtaining an optimized advanced spar-type floating platform design (Sect. 4.3).

Both utilize a framework for automated simulation and optimization developed at Fraunhofer IWES and presented in Sect. 4.2.

### 4.1 Preprocessing automated system simulations

Standardization and classification bodies, such as IEC (International Electrotechnical Commission) and DNV GL (Det Norske Veritas and Germanischer Lloyd), give recommendations on DLCs to be considered when designing floating offshore wind turbine systems. Thus, in the technical specification IEC TS 61400-3-2 (International Electrotechnical Commission, 2019b),

based on the international standard IEC 61400-3-1 (International Electrotechnical Commission, 2019a), and in the standard DNVGL-ST-0119 (DNV GL AS, 2018), building on the standard DNVGL-ST-0437 (DNV GL AS, 2016), a substantial number of DLCs is listed which cover different operating states at various environmental conditions. When performing an iterative design optimization approach, however, it is not practical to simulate the full set of DLCs for each design considered within the iterative optimization approach. This is not only for reasons of high computational effort, but also due to the fact that not

all DLCs may be relevant or design driving for the specified optimization problem.

Thus, in this work, the same approach as taken by Leimeister et al. (2020b) in another design optimization application example is adopted. In this, first, a limited number of DLCs, critical for the considered floating offshore wind turbine system and design optimization problem, is selected - a common approach in research studies (Leimeister et al., 2020b; Krieger et al., 2015; Matha et al., 2014; Huijs et al., 2013; Bachynski et al., 2013; Bachynski and Moan, 2012; Suzuki et al., 2011). For the

360 considered advanced spar-type floating offshore wind turbine system, described in Sect. 2, and the corresponding optimization problem, stated in Sect. 3, three DLCs according to IEC 61400-3-1 (International Electrotechnical Commission, 2019a) are selected (Leimeister et al., 2020b):

 – DLC 1.1 around rated wind speed (explicitly at 10.0 m/s, 11.4 m/s, and 13.0 m/s), as well as

 – DLC 1.3 and

365 – DLC 1.6, both at 8.0 m/s, 11.4 m/s (rated wind speed), and 25.0 m/s (cut-out wind speed).

These are chosen to cover highest thrust loads and corresponding system inclination and mean translational displacement at rated wind speeds, as well as maximum dynamic responses in extreme turbulent wind conditions or at severe irregular sea states, as the maximum total inclination angle, the maximum horizontal nacelle acceleration, and the mean translational motion make up three ($g_{15}$, $g_{16}$, and $g_{17}$) of the optimization constraints defined in Sect. 3.3, which need to be checked and adhered to.

From these selected three DLCs, 54 environmental conditions are defined, which correspond to 18 different environmental settings per DLC as summarized in Table 4. Thus, in each DLC turbulent wind with three different mean wind speeds and corresponding longitudinal turbulence intensity (TI) are considered. Per wind speed, six different wind seed numbers are taken into account to capture the randomness of turbulent wind. Three different yaw misalignment angles are used and combined with two seeds each to reduce the overall number of simulation cases. The irregular sea state, prevailing in all three DLCs, is specified through the significant wave height and peak period. Furthermore, each realization of the turbulent wind with a different wind seed uses as well a different wave seed to represent again the randomness of irregular waves. Finally, a current speed is specified for each wind speed.

**Table 4.** System parameters for preprocessing simulations of selected DLCs (Leimeister et al., 2020b).

| DLC | Wind conditions | | | | Sea conditions[*] | | |
|-----|-----|-----|-----|-----|-----|-----|-----|
| | Wind speed | Long. TI | Wind seed | Yaw misalignment | Sign. wave height | Peak period | Current speed |
| | 10.0 m/s | 18.34% | 1 ... 6 | -8°, 0°, 8° | 1.74 m | 6.03 s | 0.074 m/s |
| 1.1 | 11.4 m/s | 17.38% | 7 ... 12 | -8°, 0°, 8° | 1.99 m | 6.44 s | 0.084 m/s |
| | 13.0 m/s | 16.53% | 13 ... 18 | -8°, 0°, 8° | 2.30 m | 6.92 s | 0.096 m/s |
| | 8.0 m/s | 35.00% | 1 ... 6 | -8°, 0°, 8° | 1.44 m | 5.48 s | 0.059 m/s |
| 1.3 | 11.4 m/s | 26.97% | 7 ... 12 | -8°, 0°, 8° | 1.99 m | 6.44 s | 0.084 m/s |
| | 25.0 m/s | 16.68% | 13 ... 18 | -8°, 0°, 8° | 4.94 m | 10.14 s | 0.184 m/s |
| | 8.0 m/s | 20.30% | 1 ... 6 | -8°, 0°, 8° | 10.37 m | 14.70 s | 0.059 m/s |
| 1.6 | 11.4 m/s | 17.38% | 7 ... 12 | -8°, 0°, 8° | 10.37 m | 14.70 s | 0.084 m/s |
| | 25.0 m/s | 13.64% | 13 ... 18 | -8°, 0°, 8° | 10.37 m | 14.70 s | 0.184 m/s |

[*] Please notice that each realization of the turbulent wind with a different wind seed uses as well a different wave seed.

These 54 system simulations have already been performed by Leimeister et al. (2020b) with the original OC3 spar-buoy floating offshore wind turbine system and are in this study carried out with the modified reference floating system from Sect. 2.3. The simulations are executed automatically, utilizing the framework for automated simulation and optimization, which is introduced in Sect. 4.2 in more detail.

From the total simulation time of 800 s, the last 600 s (excluding any transients at the beginning) are evaluated with respect to the system performance criteria. The results, presented by Leimeister et al. (2020b), show that DLC 1.6 at rated wind speed (11.4 m/s) with wind seed number 11 and yaw misalignment angle of 8° is most critical for the total inclination angle of the system and yields the second highest value (just less than 1% lower than the maximum value obtained from all DLCs) for the horizontal nacelle acceleration. The mean translational motion is in general far off the limit value and is just less than 3.5% of the overall maximum value for the above mentioned critical DLC. For the modified advanced spar-type floating system, the five highest values for the three performance parameters and corresponding DLC simulation cases, as well as the position of the above described most critical DLC for the original OC3 phase IV floating wind turbine system are presented in Table 5.

This shows that DLC 1.6 at rated wind speed with wind seed number 11 and yaw misalignment angle of 8° is still of high criticality for the modified reference advanced spar-type floating system. It scores not the highest for the performance criteria; however, the total inclination angle of the system is almost 96% of the highest value obtained in the 54 DLC simulations, the horizontal nacelle acceleration is even almost 99% of the highest value occurring, and the mean translational motion is just less than 1% lower than the maximum value obtained.

**Table 5.** The highest values for the three performance parameters and corresponding DLC simulation cases, based on the modified reference advanced spar-type floating wind turbine system.

| Position | DLC | Wind speed | Wind seed | Yaw misalignment | Max(total inclination angle) |
|---|---|---|---|---|---|
| 1 | 1.6 | 11.4 m/s | 8 | -8° | 3.924° |
| 2 | 1.6 | 11.4 m/s | 10 | 0° | 3.876° |
| 3 | 1.6 | 11.4 m/s | 7 | -8° | 3.859° |
| 4 | 1.6 | 11.4 m/s | 11 | 8° | 3.761° |
| 5 | 1.6 | 11.4 m/s | 12 | 8° | 3.632° |

| Position | DLC | Wind speed | Wind seed | Yaw misalignment | Max(horizontal nacelle acceleration) |
|---|---|---|---|---|---|
| 1 | 1.6 | 25.0 m/s | 16 | 0° | 2.339 m/s$^2$ |
| 2 | 1.6 | 25.0 m/s | 14 | -8° | 2.322 m/s$^2$ |
| 3 | 1.6 | 8.0 m/s | 5 | 8° | 2.313 m/s$^2$ |
| 4 | 1.6 | 11.4 m/s | 7 | -8° | 2.312 m/s$^2$ |
| 5 | 1.6 | 11.4 m/s | 11 | 8° | 2.311 m/s$^2$ |

| Position | DLC | Wind speed | Wind seed | Yaw misalignment | Mean(translational motion) |
|---|---|---|---|---|---|
| 1 | 1.6 | 11.4 m/s | 9 | 0° | 19.533 m |
| 2 | 1.1 | 11.4 m/s | 9 | 0° | 19.455 m |
| 3 | 1.3 | 11.4 m/s | 9 | 0° | 19.455 m |
| 4 | 1.6 | 11.4 m/s | 12 | 8° | 19.430 m |
| 5 | 1.6 | 11.4 m/s | 8 | -8° | 19.351 m |
| 6 | 1.6 | 11.4 m/s | 11 | 8° | 19.345 m |

Thus, this DLC (DLC 1.6 at 11.4 m/s wind speed with wind seed number 11 and yaw misalignment angle of 8°) is used - as already deployed in the other first-stage design optimization application example (Leimeister et al., 2020b) - for defining the environmental conditions for the system simulations throughout the subsequent iterative optimization approach, which is specified in detail in Sect. 4.3. As, however, it is not ensured that the outcome of the DLC results comparison - based on the reference advanced spar-type floating wind turbine system - does not change for the optimized floater design, the 54

 environmental conditions will be simulated subsequent to the design optimization process and the criticality of the DLCs will be assessed again, as covered in Sect. 5.4.

## 4.2 Automated optimization framework

The preprocessing DLC simulations mentioned in Sect. 4.1, as well as the actual iterative optimization approach covered in Sect. 4.3, are executed in an automated manner by means of a Python-Modelica framework for automated simulation and optimization developed at Fraunhofer IWES (Leimeister et al., 2021, 2019).

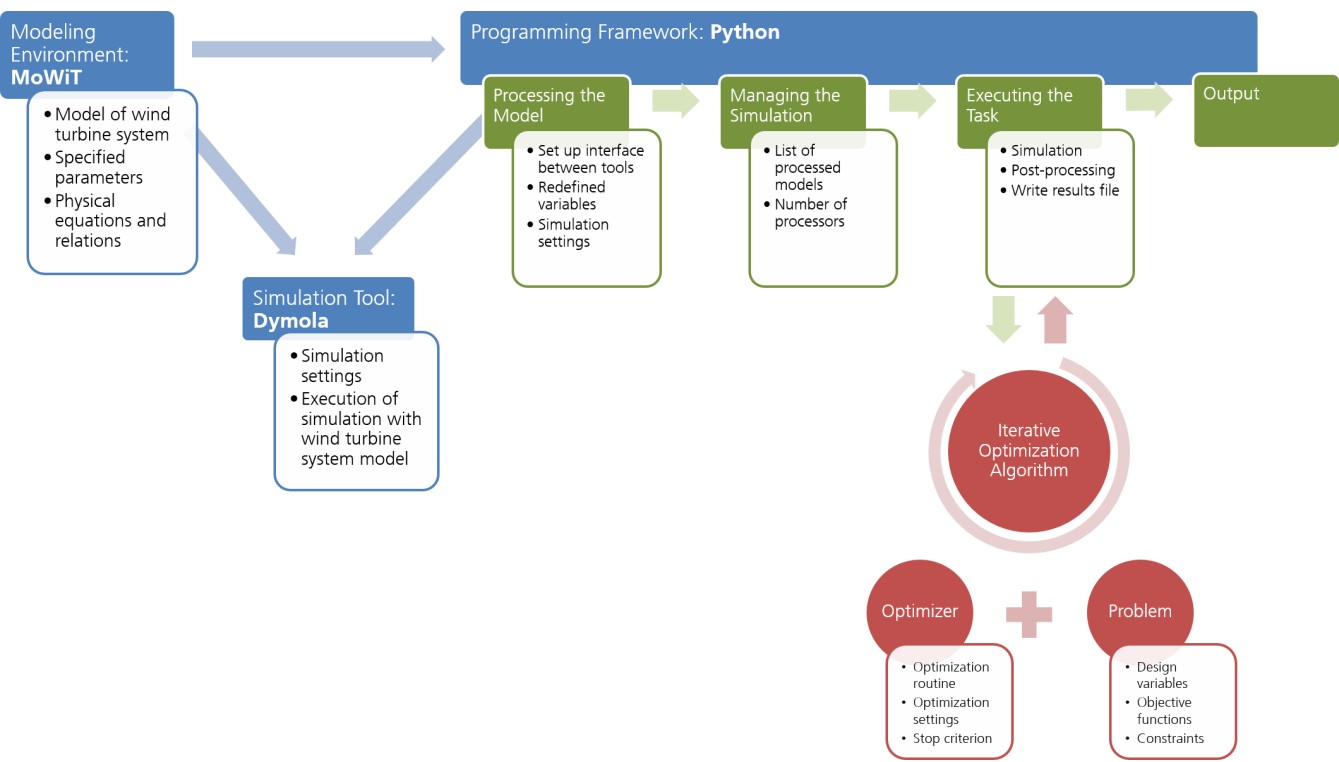

**Figure 2.** The Python-Modelica framework for automated simulation and optimization, adapted from Leimeister et al. (2021).

The structure and components of this framework are presented in Fig. 2. The framework consists of three modules: a modeling environment, a simulation tool, and a programming framework.

1. The modeling environment is MoWiT, which is already introduced in Sect. 2.3. By means of the component-based library a computational model for fully-coupled aero-hydro-servo-elastic wind turbine load calculations of the system of interest is programmed in the open-source object-oriented and equation-based modeling language Modelica. The system of interest could be any state-of-the-art onshore or offshore (bottom-fixed or floating) wind turbine. In this study, the advanced spar-type floating offshore wind turbine system, described in Sects. 2.2 and 2.3, is modeled in MoWiT. Thus, system and environmental parameters, as well as the underlying physical equations and relations are specified. From the

aero-, hydro-, control, and structural dynamic approaches available in MoWiT and covered in detail by Leimeister et al. (2020a), blade-element-momentum theory including dynamic stall and dynamic wake; linear Airy wave theory, Wheeler stretching, and MacCamy-Fuchs approach; built-in operating control; as well as modal reduced anisotropic beams for blades and rigid bodies for tower and floating structure, are utilized in this application.

2. Dymola (Dynamic Modeling Laboratory) by Dassault Systèmes[16], which is capable of time-domain simulations of complex Modelica models, is used as simulation tool. Herein, simulation and output intervals, integration settings, such as solver type, fixed integrator step size, or tolerance, as well as further specifications for translation, output, and debugging are defined.

3. The programming framework is developed in Python. The implemented scripts follow a four-step process. First, the interface between the three modules is established so that the provided wind turbine system model can be processed and new values can be assigned to system variables and simulation settings. This is for example done based on additional scripts for specifying the considered DLCs, so that for each of the 54 DLC simulations defined in Sect. 4.1 the respective environmental conditions (as presented in Table 4) are assigned to the corresponding model variables. Similar modifications of values of system variables are made within the iterative optimization algorithm, as explained in Sect. 4.3.3. In the second step, the model simulations are managed, as both parallel and successive execution is possible, depending on the user's preferences and the available processors. The main step is then the execution of the simulations, as well as additional post-processing scripts and documentation tasks. At this point also any iterative optimization algorithm, defined through the optimization problem, the optimizer, and the final optimization algorithm, (covered in Sect. 4.3) takes effect. Finally, the simulation results and any further specified results file are the output from the programming framework.

More detailed information on the Python-Modelica framework, both regarding the theory and structure, as well as its capabilities and some application examples, can be found in the publications by Leimeister et al. (2021) and Leimeister et al. (2019).

## 4.3 Specification and execution of the iterative optimization approach

As displayed in Fig. 2, the iterative optimization algorithm (Sect. 4.3.3) coupled to the Python-Modelica framework requires in addition to the model and simulation information also the definition of the optimization problem (Sect. 4.3.1) and specification of the optimizer (Sect. 4.3.2).

### 4.3.1 Optimization problem

The optimization problem comprises the specification of design variables, objective functions, as well as constraints. This is defined and described in detail in Sect. 3 and, hence, consists of seven design variables (diameters and heights of each of the three BC parts, as well as height of the ballast), one objective function for the structure material volume of the advanced

---

[16]http://www.dymola.com/ (Accessed: 4 February 2020)

spar-type floater, and 25 inequality constraints (14 for the allowable value ranges of each of the design variables, three for
the floating system performance, two for the draft requirements, and six for compliance checks regarding the filling capacity
and actual ballast height, feasible ballast densities, as well as the cylinder diameters and wall thicknesses). These are directly
implemented in the Python-Modelica framework, based on the definitions given in Sect. 3.

### 4.3.2 Optimizer

From the broad range of available algorithms and methods (Leimeister et al., 2021), only gradient-free optimization algorithms
can be chosen for the application to complex fully-coupled wind energy systems modeled by means of MoWiT. In general, for
such a complex engineering system, as a floating wind turbine is, evolutionary algorithms are highly suited to find the global
optimum of a defined optimization problem (Mishra et al., 2017). From the implemented and tested optimizers NSGAII (Non-
dominated Sorting Genetic Algorithm II), NSGAIII (Non-dominated Sorting Genetic Algorithm III), and SPEA2 (Strength
Pareto Evolutionary Algorithm 2) - all from Platypus[17] - NSGAII is found to be the most suitable optimizer for the multi-
objective optimization problem in the first-stage design optimization application example on a common floating offshore spar-
buoy wind turbine system (Leimeister et al., 2020b). As a genetic algorithm can deal with both formulations of an optimization
problem (single-objective and multi-objective) and, hence, also with the optimization problem considered in this study, which
holds only one objective function as defined in Sect. 3.2, and as the system simulations with the iterative optimization algorithm
based on NSGAII can be also parallelized in a highly efficient manner, it is stuck in this work to the well-performing - both
with respect to the convergence speed and the compliance rate concerning the constraints - optimizer NSGAII.

For the genetic algorithm NSGAII, which follows the principle of Darwin's theory of evolution - meaning having individuals
which develop further and further each generation towards performing better with respect to the fitness (objective) function -,
the number of individuals in each generation (the population size), the strategies for representing the evolution, and the stop
criterion for terminating the iterative optimization algorithm have to be defined.

– Due to the complex optimization problem with seven design variables and 25 constraints, the population size is set equal
   to the maximum possible number of processors, on which simulations can be run simultaneously. On an AMD Ryzen
   Threadripper 2990WX 32-Core Processor with 64-bit system and 64 virtual processors 60 processors could be used for
   parallel simulations. Hence, 60 individuals are considered in each generation.

– The individuals are randomly generated. When evaluating the objective function and constraints, the dominant individ-
   uals - each selected based on a comparison of two individuals - form the basis for the next generation, which is created
   without using any variator. These are the default generator, selector, and variator settings of NSGAII in Platypus.

– The stop criterion for terminating the iterative optimization algorithm is defined through the total number of simula-
   tions to be performed, while the convergence is checked separately when post-processing the simulation results. As the
   convergence speed is not known ahead of the execution of the specific optimization problem, the experience from the

---

[17]https://platypus.readthedocs.io/en/latest/ (Accessed: 6 April 2020)

first-stage design optimization application example (Leimeister et al., 2020b) is used and the total number of simulations is increased to account for the much more complex optimization problem considered in this study. Hence, the resulting number of generations being simulated is roughly tripled, so that a total number of simulations of 10,000 is chosen, corresponding to more than 166 full generations with 60 individuals each.

### 4.3.3 Optimization algorithm

Now, having defined and modeled the floating offshore wind energy system as described in Sects. 2.2 and 2.3, stating the simulation settings as given in Table 6, having specified the optimization problem (see Sects. 3 and 4.3.1), and having selected the optimizer and corresponding parameter values as outlined in Sect. 4.3.2, the iterative optimization algorithm can be executed by means of the Python-Modelica framework for automated simulation and optimization.

**Table 6.** Simulation settings.

| Simulation variable | Value | Note |
| --- | --- | --- |
| Simulation interval | from 0 s to 800 s | The first 200 s are accounted for as pre-simulation time to exclude any transients. |
| Output interval length | 0.05 s | |
| Solver | Rkfix4 | (Runge-Kutta fixed-step and 4th order method) |
| Fixed integrator step-size | 0.01 s | |

     Within the iterative optimization algorithm, the values of the design variables for the 60 individuals of the first generation
(number 0) are selected by the optimizer based on the specified allowable value ranges. All individuals are simulated in parallel on the available 60 processors and analyzed afterwards by the optimizer with respect to their fitness - meaning the objective function - and their compliance with the constraints based on the resulting time series, evaluated between 200 s and 800 s. As also simulations may have failed (due to too bad performance or instability of the considered floating wind turbine system), the simulated time is checked against the specified simulation stop time (800 s according to Table 6). In case of an unsuccessful
simulation and hence incomplete time series, the parameters of interest addressed in the constraints $g_{15}$ to $g_{17}$ for the system performance are not taken by evaluating the time series but are set equal to twice the maximum allowable value, meaning

- $\max(\text{total inclination angle})|_{\text{failing system}} = 2 \cdot 10.0° = 20.0°$

  $\Rightarrow g_{15}(system(X)|_{\text{failed}}) = 20.0° - 10.0° = 10.0° \nleq 0$

- $\max(\text{horizontal nacelle acceleration})|_{\text{failing system}} = 2 \cdot 1.962 \, \text{m/s}^2 = 3.924 \, \text{m/s}^2$

$\Rightarrow g_{16}(system(X)|_{\text{failed}}) = 3.924 \, \text{m/s}^2 - 1.962 \, \text{m/s}^2 = 1.962 \, \text{m/s}^2 \nleq 0$

- $\text{mean translational motion}|_{\text{failing system}} = 2 \cdot 64.0 \, \text{m} = 128 \, \text{m}$

  $\Rightarrow g_{17}(system(X)|_{\text{failed}}) = 128 \, \text{m} - 64.0 \, \text{m} = 64.0 \, \text{m} \nleq 0$

This way, it can be ensured that unsuccessful simulations do not comply with all constraints and, hence, are undesirable design solutions, which the optimizer then discards from further selection of well-performing individuals.

Having evaluated the simulated individuals of generation 0, the optimizer selects the design variables for the individuals of the next generation (number 1), again in accordance with the specified allowable value ranges, but also based on the fitness and constraints compliance rate of each of the previous individuals, using the tournament selector for evaluating the dominance. Then, the loop of simulating the individuals, evaluating each system with respect to the objective function and constraints, and re-selecting values (from the allowable value ranges) for the design variables of the individuals of the next generation based on the performance of the individuals in the previous generation is repeated as long as the number of executed simulations is still below the specified total number of simulations of 10,000. This iterative optimization algorithm ends when the stop criterion is reached - the final results are now available.

## 5 Results

The optimization algorithm with the specified optimization settings is executed; however, the simulation run has to be interrupted due to a required system restart. At that time already 8,133 individuals have been simulated. To complete the specified 10,000 simulations without having any disruptive effects on the final results, the optimization is continued by providing the individuals of the last wholly simulated generation 133 as start population of the subsequent optimization execution, utilizing the operator InjectedPopulation available in Platypus. Thus, the optimization run takes effectively about 31 days and eleven hours and comprises 10,011 individuals simulated in total, ranging from generation 0 up to generation 166, with full populations up to and including generation 165.

### 5.1 Developments throughout the iterative optimization process

Figure 3 shows in light blue for all simulated individuals of the optimization run the values for the design variables $x_1$ to $x_7$, as defined in Sect. 3.1. The values of the design variables of the reference advanced spar-type floating wind turbine system, covered in Sect. 2.3, are plotted additionally as red lines for comparative purposes. Post-processing of the simulation results and checking the constraints yield the dark blue recolored individuals which comply with all specified constraints. The finally selected optimum, which is presented in Sect. 5.3, is marked with a yellow filled circle framed in orange.

The developments of the design variables throughout the iterative optimization process show that in the first generations, the optimizer selects individuals covering the entire design space; however, none of the first is meeting all requirements. With more generations, the compliance rate is significantly increased, while it slightly decreases again when the focus of minimizing the objective function is coming more to the fore again. Overall, the spread in the design variables is decreased for more generations being simulated and for some design variables the change in their values is even very limited for the individuals which comply with all constraints. This indicates that the optimization algorithm is converging, though it has not yet fully converged, which is underlined by the fact that the optimum originates from the last generation.

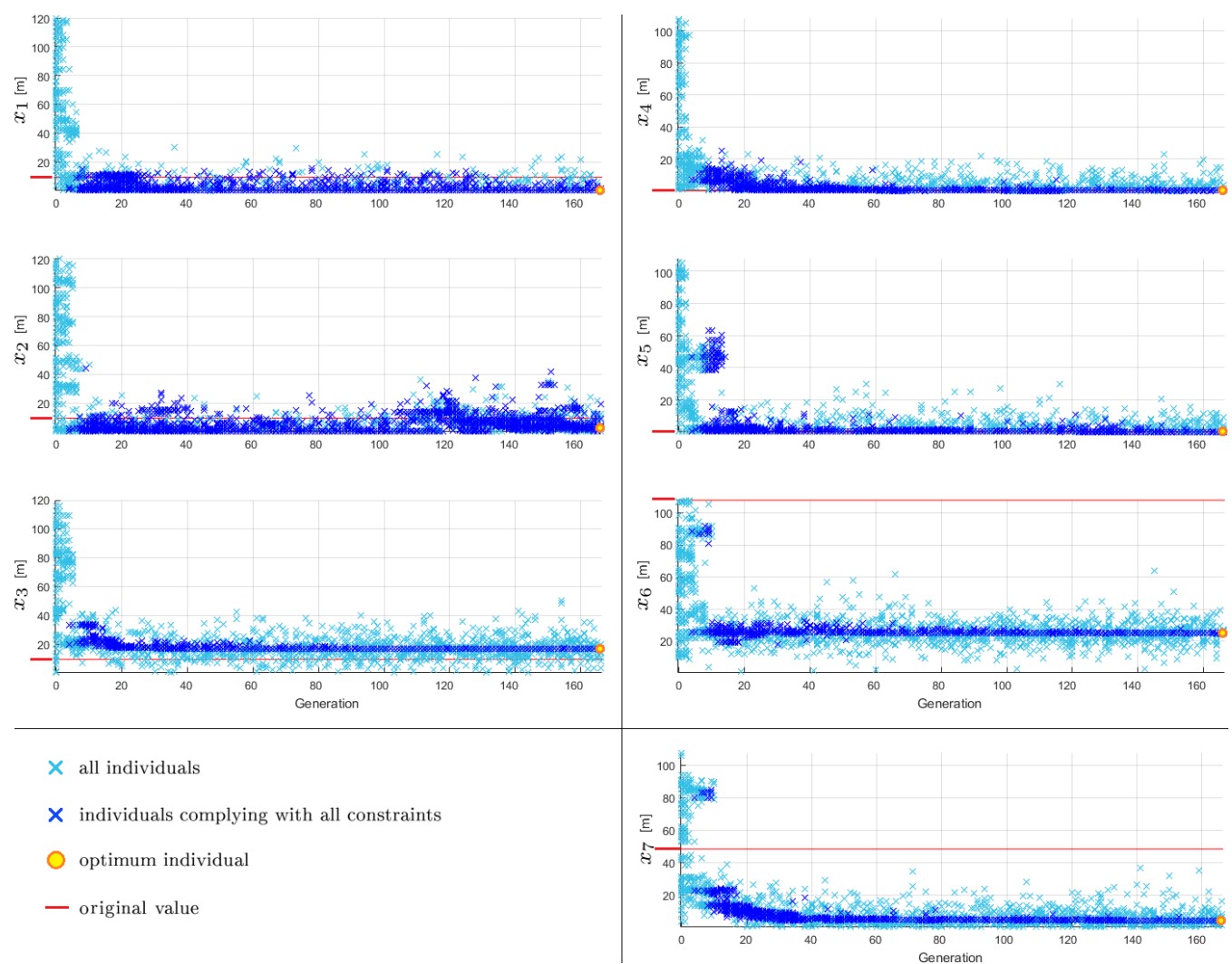

**Figure 3.** Development of the design variables throughout the iterative optimization process.

Similarly, the developments of the constraints $g_{15}$ to $g_{25}$ throughout the iterative optimization process are analyzed and presented in Fig. 4. The first 14 constraints for the allowable value ranges of the design variables are excluded, as they are not constraints that are evaluated after the simulation but are taken into account ahead of the simulations when the optimizer selects the design variables for the new individuals and, hence, are never violated. This can clearly be seen in Fig. 3, where all individuals lie within the allowable value ranges of the design variables. In Fig. 4, the light cyan crosses indicate the results for all simulated individuals, while the individuals which simultaneously comply with all constraints are recolored in dark bluish green. The limits of the inequality constraints, which should all be less or equal to zero, are indicated in red. The finally selected

optimum is marked again with a yellow filled circle framed in orange. For $g_{21}$ and $g_{22}$ it has to be noted that the ordinate is limited to [-1E+4, 1E+4] for reasons of clarity, as a few more individuals yield values in the order of magnitude of six.

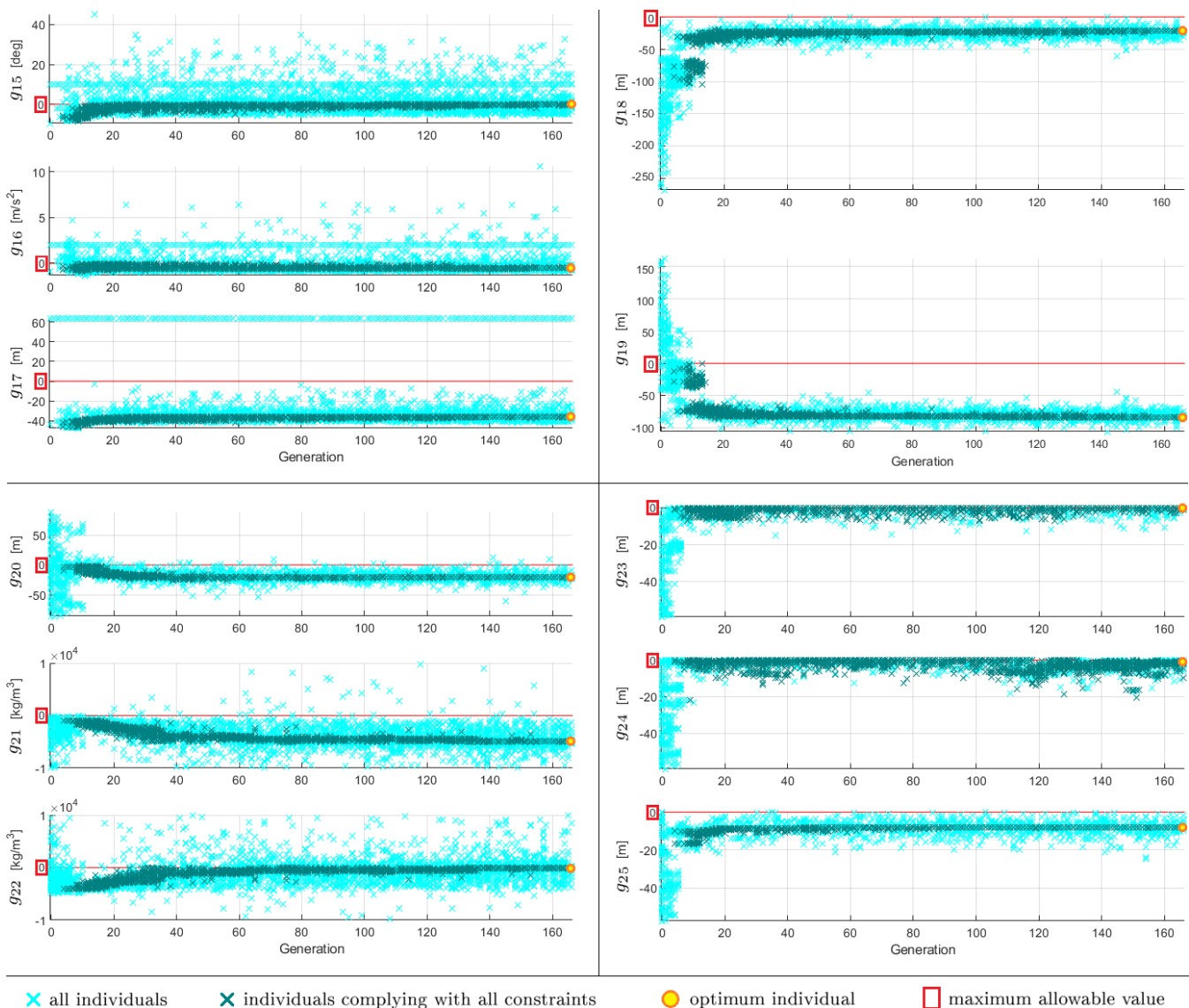

✕ all individuals    ✕ individuals complying with all constraints    ○ optimum individual    □ maximum allowable value

**Figure 4.** Development of the constraints throughout the iterative optimization process.

For $g_{18}$ to $g_{20}$ and $g_{23}$ to $g_{25}$, which are directly related to and dependent on the design variables, the developments of the constraints show a similar behavior as the developments of the corresponding design variables throughout the iterative optimization process. For the other constraints, the trend is rather different, having a large spread in the results throughout the simulated generations. The fact that for $g_{15}$ to $g_{17}$ only a few distinguishable individuals are plotted in the first generations is caused by the large number of unsuccessful simulations in the first trials of the optimizer, for which reason the performance

variables are set to the undesired values, as explained in Sect. 4.3.3, and, hence, are all the same for all failing systems. This is as well visible throughout the generations, as there is a line at the specified undesired value formed by the individuals that do
not complete the simulations successfully.

## 5.2 Advanced spar-type floater geometries in the design space

As presented and mentioned in Sect. 5.1, the individuals of the first generations cover the entire design space, specified through the allowable value ranges prescribed by means of the constraints $g_1$ to $g_{14}$. The individuals that comply with all constraints, however, are in a much more narrow area of the design space. The geometric design variables $x_1$ to $x_6$ of these individuals,
setting height and diameter of each BC part in correlation, are plotted in light blue unfilled circles in Fig. 5. The original and optimum designs are highlighted by red and blue, respectively, filled circles. From these individuals, which comply with all constraints, seven examples are selected to demonstrate the diversity of potential (meaning successful but maybe not yet optimum) advanced spar-type floater geometries. These examples are schematically drawn with black lines in Fig. 5 together with the original shape in red and having represented the ballast heights in dashed lines. The corresponding figures for design
variables, performance parameters, objective function, and further resulting geometrical and structural parameters of the presented examples are outlined in Table 7. These numbers also underline that - when evaluating $g_1$ to $g_{25}$ - none of the inequality constraints is violated.

Looking at the floater geometries presented in Fig. 5, it becomes clear that not all of these shapes can be realized with conventional structural solutions, where cylindrical sections are welded together. It has to be emphasized that these results are
560 solely based on the hydrodynamic and system-level analyses, as specified within the optimization problem. Other additional types of analyses - addressing structural integrity, manufacturability, and localized design - can, hence, deem some of the presented potential design solutions unfeasible, which is discussed in some more detail in Sect. 6. However, the advantage of this methodology - by focusing only on the hydrodynamics - is that a new range of potential floater designs is opened up and shapes like these presented in Fig. 5 can still be considered as feasible solutions when different structural realization
approaches are applied. These approaches can range from truss structures to tendons to realize large diameter changes, as well as very thin elements, without utilizing tapered sections or having issues with the structural integrity. Idea and impulse provider for such different structural realization approaches can be, for example, the oil and gas industry (Chen et al., 2017; Perry et al., 2007; Bangs et al., 2002) or innovative floating platform concepts, such as the TetraSpar by Stiesdal A/S (Stiesdal, 2019) or the pendulum-stabilized Hexafloat floater by Saipem, realized in the AFLOWT project (Richard, 2019).

## 5.3 The optimized advanced spar-type floater

Due to the single-objective nature of the optimization problem, the selection of the optimum solution happens directly through evaluating the one and only objective function. This means that from all individuals that comply with all constraints, this is chosen as optimum which exhibits the lowest value for the structure material volume of its advanced spar-type floating platform design.

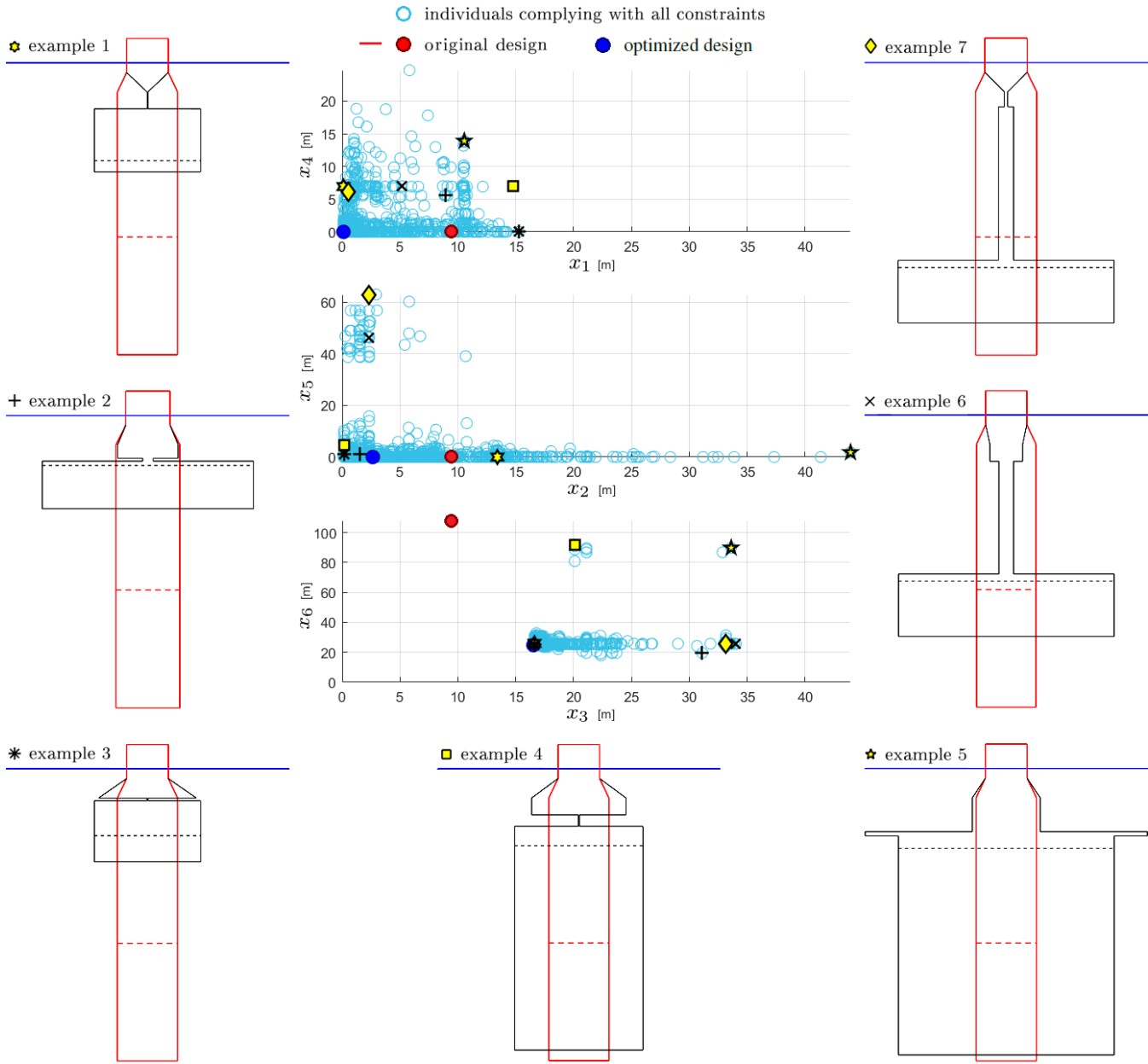

**Figure 5.** Exemplary potential advanced spar-type floater geometries selected from the individuals complying with all constraints.

First, looking at the development of the objective function $f_1$ throughout the iterative optimization process, as presented in Fig. 6, the trend of all simulated individuals (plotted in light green) shows a significant minimization of the objective function - clearly below the original value of 136.3 m³, indicated in Fig. 6 by a red line - after a large spread in the first generations.

**Table 7.** Key figures of the exemplary potential advanced spar-type floater geometries.

| Ex. | Gen. | Ind. | $x_1$ [m] | $x_2$ [m] | $x_3$ [m] | $x_4$ [m] | $x_5$ [m] | $x_6$ [m] | $x_7$ [m] | Ballast density [kg/m$^3$] | Wall thickness [m] | Draft [m] |
|---|---|---|---|---|---|---|---|---|---|---|---|---|
| 1 | 115 | 45 | 0.116 | 13.410 | 16.612 | 6.930 | 0.002 | 25.903 | 4.573 | 4.585E+03 | 0.0578 | 44.836 |
| 2 | 14 | 15 | 8.899 | 1.528 | 31.100 | 5.551 | 1.183 | 19.518 | 17.774 | 1.003E+03 | 0.1052 | 38.252 |
| 3 | 78 | 32 | 15.253 | 0.164 | 16.612 | 0.018 | 1.109 | 25.033 | 10.709 | 2.156E+03 | 0.0580 | 38.160 |
| 4 | 8 | 6 | 14.755 | 0.172 | 20.090 | 6.970 | 4.665 | 91.993 | 84.016 | 1.037E+03 | 0.0797 | 115.628 |
| 5 | 9 | 45 | 10.550 | 43.919 | 33.605 | 13.896 | 1.798 | 89.776 | 84.684 | 1.008E+03 | 0.1344 | 117.470 |
| 6 | 10 | 8 | 5.158 | 2.331 | 34.015 | 6.997 | 46.270 | 25.683 | 22.727 | 1.022E+03 | 0.1135 | 90.950 |
| 7 | 9 | 57 | 0.523 | 2.331 | 33.154 | 6.159 | 62.944 | 25.683 | 22.727 | 1.013E+03 | 0.1106 | 106.786 |

| Ex. | Max(tot. inclination angle) [°] | Max(hor. nacelle acceleration) [m/s$^2$] | Mean(transl. motion) [m] | $f_1$ [m$^3$] | Steel mass [kg] | Ballast mass [kg] |
|---|---|---|---|---|---|---|
| 1 | 9.9 | 1.337 | 28.155 | 99.1 | 7.778E+05 | 4.544E+06 |
| 2 | 5.0 | 1.231 | 22.241 | 266.2 | 2.090E+06 | 1.355E+07 |
| 3 | 9.3 | 1.724 | 27.308 | 107.7 | 8.455E+05 | 5.004E+06 |
| 4 | 2.6 | 1.955 | 17.503 | 530.1 | 4.162E+06 | 2.761E+07 |
| 5 | 1.6 | 1.664 | 21.089 | 1428.6 | 1.121E+07 | 7.570E+07 |
| 6 | 3.9 | 1.447 | 21.109 | 407.9 | 3.202E+06 | 2.111E+07 |
| 7 | 4.6 | 1.159 | 22.138 | 384.8 | 3.021E+06 | 1.987E+07 |

Zooming into the objective function results from generation 40 on, as included in Fig. 6, provides a much clearer indication of the development of the minimum structure material volume for the individuals which comply with all constraints (recolored in dark green): they aggregate to an asymptote. This is already visible in early generations; however, the spread in the objective function results of the individuals complying with all constraints is decreasing with more generations being simulated. This asymptotic clustering of the individuals which comply with all constraints to a minimum objective function value on the one hand states the convergence of the iterative optimization process and on the other hand portends that there will be several - more or less similar (elaborated in the following) - design solutions, which yield comparable low structure material volumes that are all very close to the minimum value observed.

The individual with the minimum structure material volume is pointed out in Fig. 6 by means of a yellow filled circle framed in orange. This design solution yields a reduction of the structure material volume of more than 31% compared to the original (modified) advanced spar-type floating platform. The fact that this optimum solution is just found in the last generation simulated states that full convergence is not yet reached, despite the converging trend in most of the design variables and constraints, as well as in the objective function. Nevertheless, due to the asymptotic aggregation of the individuals mentioned

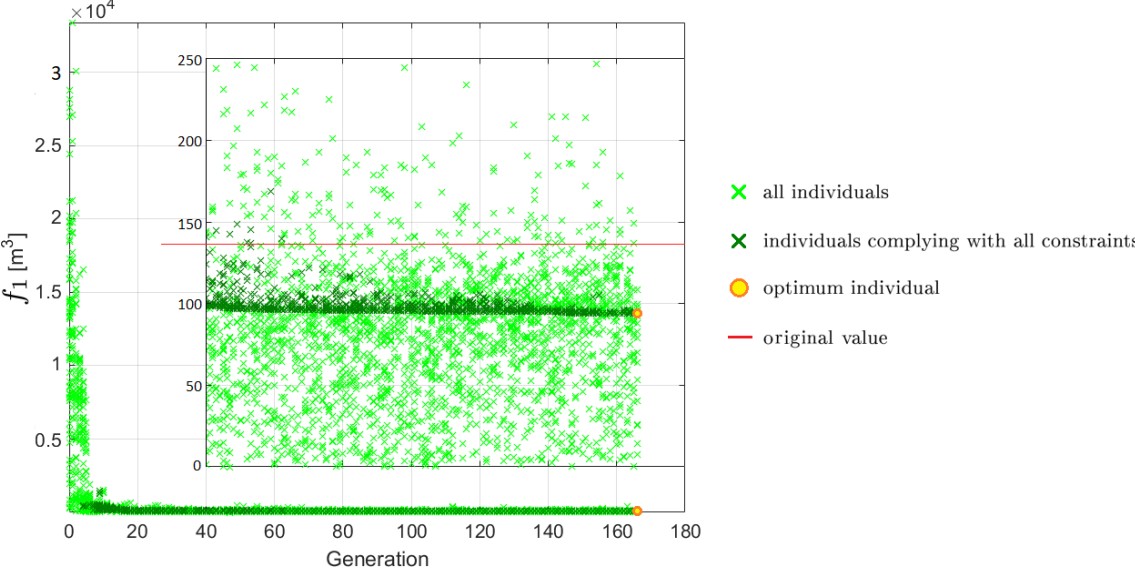

**Figure 6.** Development of the objective function throughout the iterative optimization process.

above, the first ten minimum objective function results from the individuals which comply with all constraints are evaluated. This results - as some individuals yield the same objective function value - into 16 individuals with a just by 2.84E-4% increased structure material volume, comparing the tenth lowest with the minimum value, and shapes that are difficult to distinguish from each other. This proves the above mentioned anticipation that - due to the convergence of the iterative optimization process and the aggregation of the individuals' objective function results to an asymptote - several very similar advanced spar-type floater design solutions of comparable low structure material volumes are found.

The geometry of the optimized advanced spar-type floater shape (black line) is shown schematically in Fig. 7 in comparison to the original floating platform drawn in red. The key figures of the optimized advanced spar-type floater geometry are presented in Table 8. The found design solution is - as already mentioned - out of the last generation, indicating that the optimizer is still searching for individuals with lower structure material volume; however, the improvement within the last simulated generations is negligible as outlined above. Both Fig. 7 and Table 8 indicate the following design development trend within the iterative optimization process: To reduce the structure material volume

– the overall length of the floating platform is significantly decreased compared to the original geometry - the draft of the advanced spar-type floater is, however, still significantly away from the minimum allowable draft of 15 m;

– the width of the bottom part of the support structure is enlarged, while

– the upper and middle parts are almost left out, leading to this significant constriction in the tapered part; and

– a very low ballast volume is obtained through a significantly increased ballast density, utilizing MagnaDense or high density concrete as ballast material.

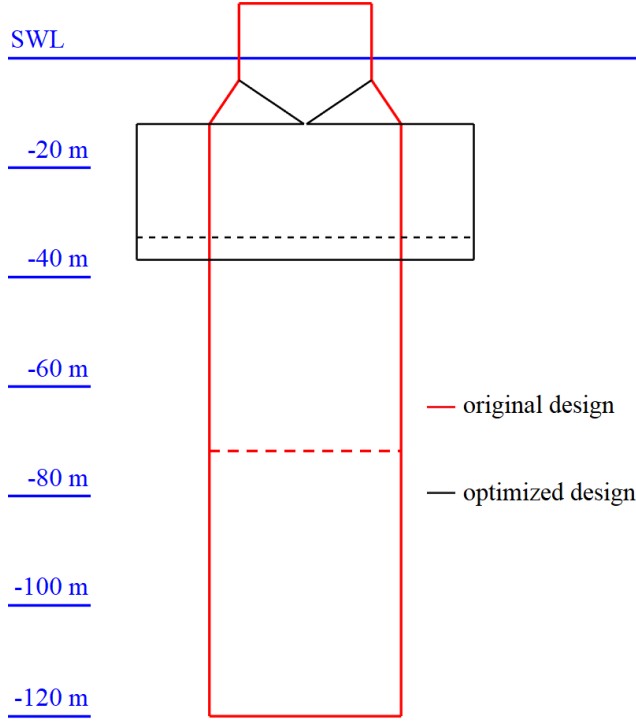

SWL

-20 m

-40 m

-60 m

-80 m

-100 m

-120 m

—— original design

—— optimized design

**Figure 7.** The optimized advanced spar-type floater geometry in comparison to the original shape.

**Table 8.** Key figures of the optimized advanced spar-type floater.

| Key figure | Value |
|---|---|
| Generation | 166 |
| Individual | 51 |
| $x_1$ | 0.115 m |
| $x_2$ | 2.653 m |
| $x_3$ | 16.525 m |
| $x_4$ | 0.001 m |
| $x_5$ | 3.0E-8 m |
| $x_6$ | 24.761 m |
| $x_7$ | 4.098 m |
| Ballast density | 4.855E+03 kg/m$^3$ |
| Wall thickness | 0.0571 m |
| Draft | 36.762 m |
| Max(tot. inclination angle) | 10.000° |
| Max(hor. nacelle acceleration) | 1.426 m/s$^2$ |
| Mean(transl. motion) | 28.394 m |
| $f_1$ | 93.9 m$^3$ |
| Steel mass | 7.373E+05 kg |
| Ballast mass | 4.267E+06  kg |

The system performance - maximum total inclination angle, maximum horizontal nacelle acceleration, and mean translational motion - points out that the maximum total inclination angle is the most critical performance criterion, as the obtained value from the optimized design is equal to the specified upper limit of $10°$.

Overall, the shape of the optimized advanced spar-type floater design resembles rather a submerged thick barge-type floater, hanging below the upper column element. This constriction in the tapered part is significant and would not directly be technically feasible, both from a manufacturing point of view and with respect to structural integrity. The reason for the current shape obtained is the connection of the upper column to the upper BC part, which, however, is, as well as the middle BC part, negligible. Thus, the tapered part could directly connect the end of the upper column with the top of the lower BC part, which is mainly purely the base column of the advanced spar-type floater. The change in required structure material would be not that significant; however, the related change in the displaced water volume has to be taken into account by adjusting the structure mass and by carefully evaluating the system performance due to the shifted center of buoyancy. This realization by means of a tapered section, however, comes with a large diameter change and corresponding large taper angle, which may be critical for both hydrodynamic load calculations and manufacturing, as discussed in more detail in Sect. 6. However, the structural issues due to the geometrical configuration of the optimized floater as presented in Fig. 7, or as well as due to the large diameter

change when utilizing a tapered section, become void when eliminating the negligible upper and middle BC parts and connecting the upper column and lower BC part by means of a number of rigid slender braces or some tendons, in combination with plated partial bulkheads for load transfer, instead of using a tapered segment. These manufacturing solutions go beyond the conventional structural realization approach of welding cylindrical sections together, but they make the found optimized floater design solution feasible and are expected to represent similar system performance. The fitness of the floater solution proposed by the optimizer is underlined due to its similarity (with respect to the innovative structural realization approach) to the most novel and alternative solutions suggested by the research community, such as the Stiesdal's TetraSpar (Stiesdal, 2019) or the Hexafloat by Saipem (Richard, 2019).

## 5.4 Performance of the optimized system in different environmental conditions

With the design solution for the advanced spar-type floating offshore wind turbine platform obtained from the optimization run, finally, the DLCs that are selected for the preprocessing automated system simulations for choosing the most critical DLC (as presented in Sect. 4.1) are rerun to check whether a shift in the most critical DLC happens. The criticality is again assessed by evaluating the fully-coupled system performance criteria (maximum total inclination angle, maximum horizontal nacelle acceleration, and mean translational motion) and analyzing the corresponding constraints $g_{15}$ to $g_{17}$. The highest values and corresponding DLC simulation cases, as well as the values obtained with the selected DLC 1.6 at rated wind speed with wind seed number 11 and yaw misalignment angle of $8°$, are presented in Table 9.

For the design solution from the optimization run, there is a shift in the criticality of the DLCs observed. The smallest change in the order of criticality of the 54 environmental conditions happens in the horizontal nacelle acceleration. Still the cases from DLC 1.6 at cut-out wind speed, as well as around rated wind speed, are most critical, but the DLC used within the iterative optimization algorithm is still among the first ten with an acceleration value that is almost 12% lower compared to the maximum obtained from all simulated DLCs. This, however, is itself still more than 17% below the maximum allowable horizontal nacelle acceleration and, hence, uncritical, which - on a side note - is not the case for the original floating spar-buoy wind turbine system. A significant increase in the resulting performance values and considerable change in the order of criticality of the environmental conditions is obtained for the mean translational motion. Here, the selected DLC for the optimization process drops down from the originally sixth position to the 22nd, while it is just 10% below the highest value achieved, which is still less than half of the maximum allowable value and, hence, again uncritical. However, the most sever shift in the criticality of the DLCs happens for the total inclination angle of the system. As indicated in Sect. 5.3, the maximum allowable value is already reached in the environmental condition considered for the optimization approach. This DLC, however, is for the obtained optimized design solution no longer prevailing but just on the 30th position, meaning that 29 other environmental conditions (mostly from DLC 1.1 and DLC 1.3, as well as some others from DLC 1.6) exceed the specified upper limit by up to more than 20%. In these environmental conditions, the floater design obtained from the optimization run would have to stop operation, while the overall system stability is not expected to be critical, as commonly much higher values for a parked floating wind turbine system in extreme environmental conditions are acceptable, such as $15°$ considered by Hegseth et al. (2020). However, to avoid reduced system availability, the occurring changed criticality of the DLCs has to be addressed

**Table 9.** The highest values for the three performance parameters and corresponding DLC simulation cases, based on the optimized advanced spar-type floating system.

| Position | DLC | Wind speed | Wind seed | Yaw misalignment | Max(total inclination angle) |
|---|---|---|---|---|---|
| 1 | 1.1 | 13.0 m/s | 18 | 8° | 12.061° |
| 2 | 1.1 | 11.4 m/s | 10 | 0° | 12.011° |
| 3 | 1.3 | 11.4 m/s | 10 | 0° | 12.011° |
| 4 | 1.1 | 11.4 m/s | 7 | -8° | 11.903° |
| 5 | 1.3 | 11.4 m/s | 7 | -8° | 11.903° |
| 30 | 1.6 | 11.4 m/s | 11 | 8° | 10.000° |

| Position | DLC | Wind speed | Wind seed | Yaw misalignment | Max(horizontal nacelle acceleration) |
|---|---|---|---|---|---|
| 1 | 1.6 | 25.0 m/s | 17 | 8° | 1.620 m/s$^2$ |
| 2 | 1.6 | 25.0 m/s | 18 | 8° | 1.618 m/s$^2$ |
| 3 | 1.6 | 25.0 m/s | 13 | -8° | 1.550 m/s$^2$ |
| 4 | 1.6 | 25.0 m/s | 16 | 0° | 1.521 m/s$^2$ |
| 5 | 1.6 | 25.0 m/s | 15 | 0° | 1.480 m/s$^2$ |
| 10 | 1.6 | 11.4 m/s | 11 | 8° | 1.426 m/s$^2$ |

| Position | DLC | Wind speed | Wind seed | Yaw misalignment | Mean(translational motion) |
|---|---|---|---|---|---|
| 1 | 1.1 | 13.0 m/s | 15 | 0° | 31.564 m |
| 2 | 1.1 | 11.4 m/s | 9 | 0° | 31.375 m |
| 3 | 1.3 | 11.4 m/s | 9 | 0° | 31.375 m |
| 4 | 1.1 | 13.0 m/s | 17 | 8° | 30.631 m |
| 5 | 1.1 | 11.4 m/s | 12 | 8° | 30.337 m |
| 22 | 1.6 | 11.4 m/s | 11 | 8° | 28.394 m |

already during the optimization - by, for example, considering safety factors for such critical and design-driving performance criteria. Alternatively or additionally, the performance in all environmental conditions can be further improved by subsequent optimization of the currently unaltered mooring system. These options are discussed in more detail in Sect. 6.

## 6 Discussion

In addition to the results presented, analyzed, and discussed in Sect. 5, more details on these results are addressed in the following and further aspects are discussed.

Based on the results and findings from the DLC simulations with the optimized advanced spar-type floating wind turbine system design, it is recommended to take some safety factors for the maximum allowable performance values into account. If the horizontal nacelle acceleration would have been exceeded in some of the 54 environmental conditions, it would not have been that critical, as a maximum allowable value of up to 0.3 times the gravitational acceleration constant - and not only 0.2 times as applied - is often accepted, as already mentioned in Sect. 2.4. The specific maximum allowable values for an operating floating offshore wind turbine system have to be provided by the turbine manufacturer or operator. Thus, maybe a higher inclination angle is still acceptable; however, if $10°$ are really the uppermost tolerated angle, a value of $8°$ or maximum $9°$ shall be used for the optimization constraint. A reduced maximum allowable total inclination angle can as well afterwards be applied in the post-processing of the results and, this way, a floater design performing well in all 54 environmental conditions can be obtained. The downside of this approach, however, is that a larger structure material volume would be required and that this design would not represent an optimized solution. A profitable option, hence, is to adjust the - currently excluded and unchanged - mooring system properties and layout design. By modifying these in a subsequent optimization task, the optimized floater design can be retained and at the same time the performance of the floating offshore wind turbine system in all considered environmental conditions improved - in this case especially the system inclination. Apart from the considered 54 environmental conditions, however, the optimized floating offshore wind turbine system design has to prove to withstand any potential environmental and operational condition during its design life. Thus, for a subsequent more realistic analysis, the entire set of DLCs recommended by standards,

– considering more realistic environmental conditions by accounting for various natural periods per considered sea state,

– capturing the low frequency dynamics of the floating wind turbine system through utilization of longer simulation times, and

– including also load cases with occurrence of a fault - such as grid loss - or with other transient loads - due to, for example, gusts - which might cause high accelerations and extreme loads,

has to be considered - at least in the pre-selection and final reassessment of the selected critical load case.

Considering the wide design space - especially the broad allowable value ranges for the structural diameters - and the extreme environmental conditions, included in the DLC simulations, some refinements in the model with respect to the hydrodynamic calculations are suggested.

– For an accurate representation of the hydrodynamic loads on the floating structure, the hydrodynamic coefficients have to be recalculated for each specific diameter. While the horizontal added mass coefficient, as well as the total inertia force, are already determined in dependency of the actual structural diameter and wave number, as the MacCamy-Fuchs approach is applied for each column element separately, the horizontal drag coefficient is currently not altered from the original value of 0.6. This is a valid assumption for large diameters already at low flow velocities; however, for small diameter structures, which can occur within the optimization algorithm, an around twice as large horizontal drag coefficient might be applicable (Clauss et al., 1992). In the vertical (heave) direction, both added mass and drag

coefficients are currently unchanged, while a vertical Froude-Krylov excitation force is considered, accounting for the difference between UC diameter and the diameter at the floater base. Especially for geometries with large diameter changes, as well as with large diameters, which can be regarded as heave plates, the hydrodynamic coefficients will differ from the original values for a continuous cylinder as the OC3 phase IV spar-buoy. Furthermore, the vertical Froude-Krylov excitation force would have to be adjusted to the specific geometry, when the lower BC part is connected by means of trusses or tendons to the upper column, to account for the differences between each upper and lower surfaces. This both - changes in the hydrodynamic coefficients in heave direction and adjusted vertical Froude-Krylov excitation force - will mainly affect the heave motion of the floating system, as well as the roll and pitch motions in some respect. With the geometry obtained from the optimization, however, it is expected to experience less strong system responses if the hydrodynamic coefficients are adjusted accordingly - which would benefit for example the system inclination - while the system responses will increase slightly if the vertical Froude-Krylov excitation force is determined accurately for the considered geometry.

– For more extreme environmental conditions with extreme waves and similar structures as obtained with the optimization run, which tend to have a large diameter directly at or close to the top of the BC, the event that the upper surface of such a large diameter cylinder becomes dry has to be accounted for when calculating the added mass and damping coefficients in order to not overestimate the heave and pitch added mass and, thus, to not underestimate the horizontal nacelle acceleration in case of more energetic sea states. Furthermore, having a horizontal surface close to the water surface - in the presented settings with a minimum distance of 12 m - could be as well critical structurally or maybe due to the impossibility of common service vessels to approach the wind turbine. However, it has to be noted that it is aimed to establish a floating platform optimized with respect to the hydrodynamics. This, then, needs to be compromised imposing other prevailing constraints, such as structural limits - as discussed later in more detail again - or accessibility, for which, for example, walk-to-work solutions with a gangway can be exploited.

– The applied MacCamy-Fuchs approach is in principle just valid for cylinders with vertical walls and not for cylinders with abrupt changes of diameters, leading to conical sections or even large horizontal surfaces anywhere along the column (the latter one, however, is considered again by means of the vertical Froude-Krylov excitation force, as discussed previously). If the MacCamy-Fuchs approach is applied to conical structures, the wave load from especially waves with low periods will be underestimated. This could be in the order of magnitude of up to 8% or 14% for a cone angle of around 6.7° or 12.2°, respectively, and could affect wave periods of 3 s to 6 s or 3.5 s to 7 s, according to investigations on a tapered bottom-fixed offshore wind turbine support structure (Leimeister, 2019). Thus, this potential underestimation of the hydrodynamic loading is mostly relevant for the environmental conditions of DLC 1.1, as well as for the below and at rated wind speed cases of DLC 1.3. For the design solution proposed in Sect. 5.3, in which the bottom end of the upper column is directly connected with the large diameter lower BC part, the taper angle would amount 32°. Any hydrodynamic calculations based on the MacCamy-Fuchs approach would no longer be meaningful if the design solution is realized by means of a solid tapered part. Thus, the alternative suggestion of having instead a number of

rigid slender braces would be favored. In order to ensure valid computation of the hydrodynamics already within the optimization approach, another constraint on the maximum taper angle shall be added, as implemented with a limit of 10° by Hegseth et al. (2020). This aspect is, however, less critical when allowing for different structural solutions, where trusses or tendons prevent any utilization of strongly tapered sections.

As addressed and discussed in Sects. 5.2 and 5.3, the geometrical configuration of the potential and optimized advanced spar-type floaters as presented in Figs. 5 and 7 may not be technically feasible from a structural integrity and manufacturability point of view, adopting the standard manufacturing solutions. For obtaining a high detail structural design, further localized analyses and assessments regarding the manufacturability have to be performed subsequently. However, structural integrity checks for buckling or stress concentration and for accounting for a realistic and adjustable base and lid thickness, which is currently just set to a fixed marginal value, can as well directly be integrated in the definition of the optimization problem. Nonetheless, based on the assumptions and focus of this study, which is on hydrodynamic and system-level analyses, a significantly improved and more cost-efficient floater can be achieved. This is as well feasible when considering different structural realization approaches, such as braces and truss structures or tendons, as already used in the oil and gas industry (Chen et al., 2017; Perry et al., 2007; Bangs et al., 2002) or utilized in innovative floater concepts (Richard, 2019; Stiesdal, 2019), instead of following purely the conventional structural approach of welding cylindrical and tapered sections together.

Finally and admittedly, for really considering an optimization of the wind turbine system cost, the ratio of CapEx (Capital Expenditure) to AEP (Annual Energy Production) or even the LCoE, which additionally takes OpEx (Operational Expenditure) - and sometimes also costs of decommissioning - into account, would have to be considered to be minimized. This way, a real trade-off between saved material costs, changed expenditure of manufacturing and maintenance of the system, and different system performance, and, hence, affected AEP can be found. However, this requires a more holistic and complex approach, considering annual environmental distributions at the location of interest, calculations for the full life-time of the system, as well as knowledge of possible manufacturing processes and related costs. The present work can be further expanded in the future to take into account these steps and aspects.

## 7 Conclusions

In this paper, an automated optimization approach is applied to a floating offshore wind turbine system in order to design an advanced spar-type floating platform, which is optimized with respect to the change in hydrodynamics and their impact on the main system performance, while structural, manufacturability, or other constraints are not considered. This approach, following a freer optimization formulation, is taken in order to be able to explore novel design spaces which can be better from a hydrodynamic point of view, but may require novel structural approaches, as actively investigated by the community (e.g. Stiesdal's TetraSpar and Saipem's Hexafloat). The application is based on the OC3 phase IV spar-buoy floating offshore wind turbine system. This, however, is modified by dividing the spar-buoy base column into three distinct partitions, so that sufficient buoyancy, as well as a deep center of gravity can be obtained. Furthermore, the wall thickness is adjusted based on a common ratio of the support structure's structural mass to the displaced mass of water. The optimization focuses on the minimization of

the steel volume of the floater, which represents an approximation of the CapEx of the support platform. In addition, constraints regarding the outer dimensions (meaning the allowable value ranges of the design variables), the global fully-coupled system performance, the system draft, the ballast, and the geometric integrity are defined. Having selected, based on preprocessing automated system simulations, one DLC which is most critical for the constrained system performance criteria, the iterative optimization algorithm run is performed, utilizing the Python-Modelica framework for automated simulation and optimization, as well as using the genetic algorithm NSGAII as optimizer. The analysis of the optimization simulation results shows that the individuals which comply with all prescribed constraints aggregate as for their objective function values to an asymptote. The results from the optimization run emphasize the complexity of the optimization problem and indicate that - despite the large number of simulations and the asymptotic clustering to a minimum objective function - full convergence is not yet obtained. Nevertheless, the applied iterative optimization algorithm presented in this study yields an advanced spar-type floating support structure design, which

- has a by more than 31% reduced structure material volume compared to the original floating platform,

- meets all global performance criteria for the considered critical DLC,

- has an overall draft of 36.8 m,

- utilizes MagnaDense or high density concrete as ballast material, and

- resembles a submerged thick barge-type floater.

The operability is - taking the maximum allowable system performance values as strict obligation for operating ability - limited to 46.3% of the considered 54 environmental conditions. This, however, can be much more extended when modifying subsequently the currently unchanged mooring system properties and layout. Based on the applied hydrodynamic and system-level analyses an optimized initial advanced spar-type floater design is obtained, which has to be further refined by incorporating structural checks into the optimization process, but can be realized by means of innovative structural approaches, which utilize trusses or tendons instead of solely welding cylindrical sections together.

*Author contributions.* ML: Conceptualization, Data curation, Methodology, Software, Validation, Formal analysis, Investigation, Project administration, Visualization, Writing - original draft, Writing - review; MC: Conceptualization, Methodology, Supervision, Writing - review & editing; AK: Supervision, Writing - review & editing, Funding acquisition.

*Competing interests.* The authors declare that they have no conflict of interest.

*Acknowledgements.* This work was partially supported by grant EP/L016303/1 for Cranfield University, University of Oxford and University of Strathclyde, Centre for Doctoral Training in Renewable Energy Marine Structures - REMS (http://www.remscdt.ac.uk/) from the UK Engineering and Physical Sciences Research Council (EPSRC) and the German Fraunhofer Institute for Wind Energy Systems (Fraunhofer IWES).

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
