# Peer review of "A fully integrated optimization framework for designing a complex geometry offshore wind turbine spar-type floating support structure"

_Wind Energy Science, 2020_

## Referee Comment (RC1) · Anonymous Referee #1 · 15 Jul 2020

[11pt]article [a4paper,total=6in,9in]geometry

[Figure]

**Reviewer's comments**

July 15, 2020

General comment: This paper carried out a single-objective gradient-free optimization of three-section spar-buoy floater for floating wind turbine, where the mooring stiffness is kept constant. The study is well performed and the paper is well written. However, the results are not presented in the most effective way. Further, the way to handle the mooring system needs to be improved, or verified after the optimization.

To improve the quality of the paper, a number of questions, suggestions and comments are provided below.

1. page 1, line 45. The author does not provide an adequate literature review of the current state of the art in optimization of floating wind turbine support structures, except listing eight papers. The authors should, the studies related to single objective optimization, gradient-free optimization, and spar-buoy floater, which are most relevant to the study in this paper. Besides, there are also studies of multi-objective GA optimization of floating wind turbine support structures, which are also relevant to this study. Additionally, how the mooring system is treated in the relevant studies? After an adequate literature review, the authors need to justify the value and contribution of this work.

2. page 7, line 190. A general comment is related to the assumption that the mooring system is kept constant in this study. the mooring system is composed of

a few mooring lines. Did the authors use constant values for the horizontal and vertical stiffness of each mooring line? Or, did the authors use a constant mooring stiffness matrix for the entire mooring system? The former approach is more reasonable, because the floater pitch stiffness depends on the product of the horizontal stiffness of mooring line and the radius of the fairlead. Can the authors predict what is the impact of their assumption on the optimized designs? The optimizer may take advantage of the assumption. Can the authors improve the way to treat the mooring system? This minor improvement can provide a more realistic way to include the mooring system. Alternatively, the authors may consider provide a representative design of the mooring system that satisfies the mooring stiffness for the chosen optimized design. Such practice and guide would make the methodology in the study more convincing.

3. page 17, section 4.3.2. The authors classify the optimizers into single-objective optimizers and multi-objective optimizers. It is a little confusing. While single-objective and multi-objective optimization are widely used, this often points to the formulation of the optimization problem, rather than the optimizer. The performance of the optimizer highly depends on the algorithm itself. On the other hand, for example, GA can be used to solve both single-objective and multi-objective optimization problem as stated by the authors. In a strict way, GA can be called sing-objective and multi-objective optimizer. The authors may re-write this paragraph to avoid the confusion and directly highlight that they are using GA algorithm.

4. page 30, line 658. This study lacks a verification of the optimized design. Can the authors verify the hydrodynamic properties of the floater by using high-fidelity tools such as WAMIT?

5. page 32, line 725. This study assumes a rigid floater with a constant thickness. However, the chosen final design has a neck-like weak feature. The authors

noted in the conclusion that this can be manufactured by using truss structures. Can the authors further illustrate this? Further, how would this bias the cost and performance of the chosen design?

6. A general comment is related to the computation time for the optimization problem. How long does it takes? Can the authors provide such information?

7. Another general comment is related to the interpretation of the optimized design. The authors have noted its similarity with TetraSpar. Can the authors compare the system properties of the baseline design and the optimized design? For example, the buoyancy and mass centers of the entire wind turbine, the eigen-frequencies of the coupled floater-tower vibration mode?

A few minor comments are also listed as follows:

1. page 10, section 3. It is better to modify the formulation of the optimization problem into a single-objective optimization, which is the case in this study.

2. page 10, section 3.1. It may be easier to follow, if the design variables are replaced with $d_i$ and $h_i$. Alternatively, one can also use $d_u$, $d_m$, $d_l$, $h_u$, $h_m$, $h_l$, $h_b$. But it does not affect the results. It is up to the authors.

3. page 11, line 305. "It is not practical to simulate ... the full set of DLCs". It is better to put "the full set of DLCs" right after "simulate".

4. page 11, line 307. "... might be relevant and driving the design ...". It may be changed to "... may be relevant or design driving ..."

5. page 17, line 407-412. The sentence is too long. It can be divided into three sentences.

6. page 23, Fig. 5. It is better to remove the baseline design. The text in the legend "original desing" may be "original design". The text "optimum individual" means the final chosen optimized design, which may not be the global optimum. "optimum individual" may be replaced with "optimized design".

7. page 26, Fig. 7. It is better to put the baseline design and the optimized design side by side. Then it is clearer to see the difference between the two designs.

8. page 31, line 673, "where trusses or tendons prevent any utilization of strongly tapered sections". Do the authors want to mean that the trusses or tendons support the use of strongly tapered sections?

9. page 32, line 725. The sentence is too long.

---

## Referee Comment (RC2) · Anonymous Referee #2 · 16 Jul 2020

This paper presents a geometry re-design of the OC3 spar using a genetic algorithm. To make the problem more manageable, the authors make a number of simplifying assumptions in their analysis:

- No structural analysis

- Frozen mooring design and performance (despite large changes in the geometry)

- Frozen hydrodynamic coefficients (despite large changes in the geometry)

- Relatively few design variables and constraints

[Figure]

There are some weaknesses in the methods and results that could be improved to make a stronger paper. For instance, due to these simplifying assumptions, the optimized design geometries are quite surprising and raise as many questions as answers. The authors acknowledge that these are more qualitative and instructive design geometries than immediately applicable, but in that case more sensitivity studies and trade-off studies should be executed. Also, the presentation of the methods and results is fairly long winded and somewhat repetitive. Efforts could be made to tighten up the language and organization. However, instead of discussing these weaknesses in more detail, I am more concerned with the uniqueness and level of contribution of this paper.

If I do a literature search on the keywords "floating spar optimization", I get many hits and papers going back at least 15 years, only some of which are mentioned by the authors. Some of these papers also build on the OC3 spar that the authors have chosen for their baseline and/or use genetic algorithms to explore the design space as is done here. Furthermore, many of these papers do not make the same simplifying assumptions as this work does, leaving me to think that I should trust those other papers more. This also leads me to wonder what the novel contribution to the literature here is. I do not see that clearly stated in the paper.

An even more significant concern for me is the similarity between this paper and a previous one already published by the authors that also does a similar optimization of the OC3 spar with a GA: https://doi.org/10.1016/j.oceaneng.2020.107186. Much of the material here on the methods and discretization is nearly identical to their previous paper and leaves me to wonder why these two efforts were not combined. To me, this submission has not done enough to separate itself from the authors' prior work and perhaps also not enough to separate itself from the prior work of others. What is different just doesn't meet the bar of its own journal paper, so perhaps a conference setting would be more appropriate. I am willing to hear the authors retort to my concerns, but I am inclined to decline this submission.

---

## Editor Comment (EC1) · Katherine Dykes (Editor) · 28 Jul 2020

An additional review was submitted after the review period closed. See attached file.
* * *
[Figure]

[Figure]

[Figure]

**A fully integrated optimization framework for designing a complex geometry offshore wind turbine spar-type floating support structure**

Mareike Leimeister[1,2], Maurizio Collu[1], and Athanasios Kolios[1]

[1]Naval Architecture, Ocean and Marine Engineering, University of Strathclyde, 100 Montrose Street, Glasgow G4 0LZ, United Kingdom
[2]Division System Technology, Fraunhofer IWES, Institute for Wind Energy Systems, Am Luneort 100, 27572 Bremerhaven, Germany

**Correspondence:** Mareike Leimeister (mareike.leimeister@iwes.fraunhofer.de)

**Abstract.** Spar-type platforms for floating offshore wind turbines are considered suitable for commercial wind farm deployment. To reduce the hurdles of such floating systems to become competitive, a fully integrated optimization framework is applied to design an advanced spar-type floater for a 5 MW wind turbine. Three cylindrical sections with individual diameters and heights, as well as the ballast filling height are the modifiable design variables of the optimization problem. Constraints
5   regarding the geometry, ballast, draft, and system performance are specified. The optimization objective to minimize the floater structural material shall represent the overall goal of cost reduction. Preprocessing system simulations are performed to select a critical design load case, which is used within the iterative optimization algorithm. This itself is executed by means of a fully integrated framework for automated simulation and optimization and utilizes a genetic algorithm. The presented design optimization example and approach emphasize the complexity of the optimization problem and lead to the recommendation
10   to consider safety factors for other more critical and design-driving performance criteria. For the applied methodology and conditions it is shown that the required material for an advanced spar-type platform supporting an offshore wind turbine can be reduced by more than 31% and, at the same time, the performance of the floating system - expressed by the maximum system inclination, maximum tower top acceleration, and mean translational motion - improved in some respect.

**Abbreviations:** AEP, Annual Energy Production; ALPSO, Augmented Lagrangian Particle Swarm Optimization; BC, Base Column; BC$_{low}$, Base Column lower part; BC$_{mid}$, Base Column middle part; BC$_{up}$, Base Column upper part; CapEx, Capital Expenditure; COBYLA, Constrained Optimization BY Linear Approximation; DLC, Design Load Case; DNV GL, Det Norske Veritas and Germanischer Lloyd; Dymola, Dynamic Modeling Laboratory; IEC, International Electrotechnical Commission; IWES, Institute for Wind Energy Systems; LCoE, Levelized Cost of Energy; MoWiT, Modelica for Wind Turbines; NREL, National Renewable Energy Laboratory; NSGAII, Non-dominated Sorting Genetic Algorithm II; NSGAIII, Non-dominated Sorting Genetic Algorithm III; OC3, Offshore Code Comparison Collaboration; OC4, Offshore Code Comparison Collaboration Continuation; OpEx, Operational Expenditure; Rkfix4, Runge-Kutta fixed-step and 4th order method; SPEA2, Strength Pareto Evolutionary Algorithm 2; SWL, Still Water Level; TI, Turbulence Intensity; TP, Tapered Part; UC, Upper Column

**Fig. 1.**

[Figure]

---

## Editor Comment (EC2) · Katherine Dykes (Editor) · 28 Jul 2020

An additional review was completed after the review period ended. See attached file.

Please also note the supplement to this comment:
https://wes.copernicus.org/preprints/wes-2020-93/wes-2020-93-EC2-supplement.pdf
* * *

---

## Author Comment (AC1) · 3 Aug 2020

The authors would like to thank the reviewer for the constructive recommendation and comments which will help improve the current and future work. In the following, the authors would like to respond to the reviewer's comments. The addressed comments are included in italic font.

1. *page 1, line 45. The author does not provide an adequate literature review of the current state of the art in optimization of floating wind turbine support structures, except listing eight papers. The authors should, the studies related to single objective optimization, gradient-free optimization, and spar-buoy floater, which are most relevant to the study in this paper. Besides, there are also studies of multi-objective GA optimization of floating wind turbine support structures, which are also relevant to this study. Additionally, how the mooring system is treated in the relevant studies? After an adequate literature review, the authors need to justify the value and contribution of this work.*

    The authors extend this paragraph and add a more detailed literature review on optimization applications of spar-type floating wind turbine support structures. The separate aspect of optimizing the mooring system is now included and addressed in more detail in Section 2.3 (paragraph in lines 188-195 on page 7).

2. *page 7, line 190. A general comment is related to the assumption that the mooring system is kept constant in this study. the mooring system is composed of a few mooring lines. Did the authors use constant values for the horizontal and vertical stiffness of each mooring line? Or, did the authors use a constant mooring stiffness matrix for the entire mooring system? The former approach is more reasonable, because the floater pitch stiffness depends on the product of the horizontal stiffness of mooring line and the radius of the fairlead. Can the authors predict what is the impact of their assumption on the optimized designs? The optimizer may take advantage of the assumption. Can the authors improve the way to treat the mooring system? This minor improvement can provide a more realistic way to include the mooring system. Alternatively, the authors may consider provide a representative design of the mooring system that satisfies the mooring stiffness for the chosen optimized design. Such practice and guide would make the methodology in the study more convincing.*

    The realization of the mooring system and the use of the resulting mooring system properties follows the first approach mentioned by the reviewer. Each mooring line is specified through its length, diameter, mass in water, extensional stiffness, added mass coefficient, drag coefficient, damping coefficient for inner damping, fixation point at anchor, as well as fixation point at fairlead. As mentioned (line 189 on page 7) the mooring system itself can cover a separate optimization. This is now underlined in more detail by added literature (as indicated in 1. as well). These literatures also confirm that, following the applied approach, a corresponding mooring system design, which represents the same resulting mooring system properties but matches possible attachment points on the optimized floater geometry, can be obtained through a separate subsequent optimization. The literature as well emphasizes the mentioned aspect (lines 192-195) that the system performance can further be improved through a subsequent optimization of the mooring system.

3. *page 17, section 4.3.2. The authors classify the optimizers into single-objective optimizers and multi-objective optimizers. It is a little confusing. While single-objective and multi-objective*

*optimization are widely used, this often points to the formulation of the optimization problem, rather than the optimizer. The performance of the optimizer highly depends on the algorithm itself. On the other hand, for example, GA can be used to solve both single-objective and multi-objective optimization problem as stated by the authors. In a strict way, GA can be called sing-objective and multi-objective optimizer. The authors may re-write this paragraph to avoid the confusion and directly highlight that they are using GA algorithm.*

> The paragraph is reformulated, to ensure that the currently termed single-objective and multi-objective optimizers mean optimizers that can deal with single-objective and multi-objective formulated optimization problems.

4. *page 30, line 658. This study lacks a verification of the optimized design. Can the authors verify the hydrodynamic properties of the floater by using high-fidelity tools such as WAMIT?*

> Unfortunately, the authors do not have a license to other high-fidelity tools, such as WAMIT, for performing a verification of the specific optimized design. The hydrodynamic properties and calculations performed within MoWiT are verified for other geometries (OC3 spar-buoy, OC6 semi-submersible, but also a large diameter bottom-fixed monopile or the OC5 jacket), where data from other tools for comparison was available. As pointed out on page 30, the proposed realization of the optimized spar-buoy floater design without having that strongly constricted shape or instead of this using a tapered connection between the upper column and the bottom part of the base column, but with utilizing tendons for connecting the bottom part of the base column to the upper column, will not experience the shortcomings of the hydrodynamic calculation approaches.

5. *page 32, line 725. This study assumes a rigid floater with a constant thickness. However, the chosen final design has a neck-like weak feature. The authors noted in the conclusion that this can be manufactured by using truss structures. Can the authors further illustrate this? Further, how would this bias the cost and performance of the chosen design?*

> The innovative structural realization opportunities are explained in lines 513-520 on pages 22/23, lines 571-579 on page 27, and lines 679-683 on page 31. Here it is meant that the bottom part of the base column can be connected to the upper column by means of tendons or truss elements. Thus, it is not meant that the optimized spar-buoy geometry is fully replaced by a truss structure, but instead of having tendons between upper column and bottom part of base column, also rigid braces/truss elements might be used. As mentioned in lines 575-577 on page 27, it is expected that such an alternative structural realization – if it represents a rigid connection – will represent similar system performance. With respect to the costs, it might be more comparative to use tendons instead of truss elements, however, this would imply a more detailed analysis including manufacturing costs in addition to material costs.

6. *A general comment is related to the computation time for the optimization problem. How long does it takes? Can the authors provide such information?*

> The information on the computation time has already been provided in line 463 on page 19.

7.  *Another general comment is related to the interpretation of the optimized design. The authors have noted its similarity with TetraSpar. Can the authors compare the system properties of the baseline design and the optimized design? For example, the buoyancy and mass centers of the entire wind turbine, the eigen-frequencies of the coupled floater-tower vibration mode?*

    The authors reformulate the statement to ensure that the similarity of the optimized spar-buoy floater with TetraSpar is purely meant with respect to the innovative structural realization approach and not referring to the specific system properties.

Thanks as well for the minor comments added. Even if the reviewer leaves for some points the final decision on the implementation to the authors, the authors also would like to respond to these comments.

1.  *page 10, section 3. It is better to modify the formulation of the optimization problem into a single-objective optimization, which is the case in this study.*

    By setting $l = 1$, as done in Section 3.2 on page 11, the prevailing case of a single-objective optimization problem is defined.

2.  *page 10, section 3.1. It may be easier to follow, if the design variables are replaced with di and hi. Alternatively, one can also use du, dm, dl, hu, hm, hl, hb. But it does not affect the results. It is up to the authors.*

    The authors prefer to follow the general formulation of an optimization problem with design variables $x_i$, objective functions $f_i$, equality constraints $h_i$, and inequality constraints $g_i$. Thus, and as the definition of the design variables $x_i$ is clearly given in Section 3.1, the authors stay with the used terms $x_i$.

3.  *page 11, line 305. "It is not practical to simulate ... the full set of DLCs". It is better to put "the full set of DLCs" right after "simulate".*

    The sentence is reordered accordingly.

4.  *page 11, line 307. "... might be relevant and driving the design ...". It may be changed to "... may be relevant or design driving ..."*

    The sentence is adjusted accordingly.

5.  *page 17, line 407-412. The sentence is too long. It can be divided into three sentences.*

    Due to the adjustments made based on the reviewer's main comment number 3, the long sentence referred to in this comment is no longer existing.

6.  *page 23, Fig. 5. It is better to remove the baseline design. The text in the legend "original desing" may be "original design". The text "optimum individual" means the final chosen optimized*

*design, which may not be the global optimum. "optimum individual" may be replaced with "optimized design".*

> The authors intend by plotting the baseline design to allow an easier and faster (visual) comparison of the presented example designs, as this way it is shown that always a similar scale is presented and the example geometries can always be put in relation to the one and the same baseline design. The text in the legend is adjusted according to the comments.

7. *page 26, Fig. 7. It is better to put the baseline design and the optimized design side by side. Then it is clearer to see the difference between the two designs.*

> For the authors it is rather easier to compare the designs and clearer to see the differences in both heights and diameters, when having both geometries plotted in one picture and having the geometries distinguished through using different colors.

8. *page 31, line 673, "where trusses or tendons prevent any utilization of strongly tapered sections". Do the authors want to mean that the trusses or tendons support the use of strongly tapered sections?*

> By means of this sentence it is meant that by allowing for alternative and innovative structural realization approaches, such as the use of tendons or truss elements, a strongly tapered section, which would be required when just following the common structural realization approach, would no longer be required.

9. *page 32, line 725. The sentence is too long.*

> This is the automatically generated author contributions statement, as required by the format of the journal.

---

## Author Comment (AC2) · 3 Aug 2020

The authors would like to thank the reviewer for the constructive recommendation and comments which will help improve the current and future work. In the following, the authors would like to respond to the reviewer's comments. The addressed comments are included in italic font.

1. *There are some weaknesses in the methods and results that could be improved to make a stronger paper. For instance, due to these simplifying assumptions, the optimized design geometries are quite surprising and raise as many questions as answers. The authors acknowledge that these are more qualitative and instructive design geometries than immediately applicable, but in that case more sensitivity studies and trade-off studies should be executed. Also, the presentation of the methods and results is fairly long winded and somewhat repetitive. Efforts could be made to tighten up the language and organization. However, instead of discussing these weaknesses in more detail, I am more concerned with the uniqueness and level of contribution of this paper.*

   On the first view, the presented approach might seem to follow more simplifying assumption than other studies on optimizing spar-buoy floating wind turbine support structures. However, as outlined in the introduction (lines 49-51 and 56-59 on page 3) and discussed in more detail throughout the paper (lines 199-203 on page 7, 513-520 on pages 22/23, lines 571-579 on page 27, and lines 679-683 on page 31, lines 695-698 on page 31) the less restricted optimization problem is chosen well-considered and deliberately, to allow the consideration of novel design solutions, including alternative manufacturing approaches and structural realization methods. The common optimization approaches and defined optimization problems consider spar-type structures, which are manufactured by welding cylindrical or tapered elements together. Due to the critical aspect of having large taper angles (as addressed in lines 658-673 on pages 30/31), as well as based on the structural aspects for the commonly manufactured floater designs, the range of potential optimized floater designs and shapes is limited. However, the manufacturing solution, such as choosing between welded conical sections and tendons or truss elements for connections, cannot directly be implemented in an optimization approach. Thus, the final solution has to be selected subsequent to the optimization. But in order to not prevent innovations, thus, this paper addresses an approach to be more open-minded and allow for alternative manufacturing approaches and structural realization methods. Thus, more design variables are defined and the corresponding allowable value ranges are specified well thought out to include the aspect of innovativeness. This approach shows a novel contribution to the future design development of floating offshore wind turbine support structures. The novelty and innovativeness of the proposed approach and resulting optimized design is substantiated by the fact that the potential structural realization approaches resemble the highly innovative concepts followed in research projects (AFLOWT, TetraSpar).

2. *If I do a literature search on the keywords "floating spar optimization", I get many hits and papers going back at least 15 years, only some of which are mentioned by the authors. Some of these papers also build on the OC3 spar that the authors have chosen for their baseline and/or use genetic algorithms to explore the design space as is done here. Furthermore, many of these papers do not make the same simplifying assumptions as this work does, leaving me to think*

*that I should trust those other papers more. This also leads me to wonder what the novel contribution to the literature here is. I do not see that clearly stated in the paper.*

> The authors add a more detailed literature review (paragraph from lines 44 to 59 on pages 2 and 3) on optimization applications of spar-type floating wind turbine support structures and point out the differences of the followed approach presented in this paper compared to other approaches found in the literature. This underlines the novel contribution of this paper, as answered also in detail in comment number 1. Thus, this freer optimization formulation of the project can allow out of the box thinking and potentially push for more disruptive designs, which can unlock the potential of floating wind.

3. *An even more significant concern for me is the similarity between this paper and a previous one already published by the authors that also does a similar optimization of the OC3 spar with a GA: https://doi.org/10.1016/j.oceaneng.2020.107186. Much of the material here on the methods and discretization is nearly identical to their previous paper and leaves me to wonder why these two efforts were not combined. To me, this submission has not done enough to separate itself from the authors' prior work and perhaps also not enough to separate itself from the prior work of others. What is different just doesn't meet the bar of its own journal paper, so perhaps a conference setting would be more appropriate. I am willing to hear the authors retort to my concerns, but I am inclined to decline this submission.*

> This paper built on, but has a completely separate aim from the previous publication by the authors (https://doi.org/10.1016/j.oceaneng.2020.107186). In the previous publication, the basic three design variables (one column with a height and diameter, as well as the ballast density), common to define a standard spar-buoy floater, are selected and the considered allowable value ranges follow directly the focus of reducing costs, material, and outer dimension, while the cost itself is not specified as objective. The work presented in that previous publication deals with a very simple structure and related simple optimization approach and also mainly deals to prove the validity of the applied optimization framework and approach. This paper submitted now to WES is substantially different and significantly advances the work already completed by the authors. This paper considers a more realistic geometry, which implies as well different methods and approaches (other and more design variables, not cost-driven but broader and well thought out selected allowable value ranges, formulation of the optimization problem directly for cost reduction, open-minded approach allowing for alternative structural realization methods and innovation). Based on this paper and applied approach, the way towards more realistic analyses, with more DLCs included and coupled with structural analyses, is as well paved.

---

## Author Comment (AC3) · 3 Aug 2020

As the comments were included in EC2 these are as well addressed in the reply to EC2.
* * *

---

## Author Comment (AC4) · 3 Aug 2020

The authors would like to thank the reviewer for the constructive recommendation and comments which will help improve the current and future work. In the following, the authors would like to respond to the reviewer's comments. The addressed comments are included in italic font.

1. *page 2, lines 35-37. It may be worth noting that the Fukushima Forward project in Japan tested together three different wind turbines connected to the same floating substation. It was the first floating wind farm, but with only a limited operating life as units were prototypes.*

   This aspect is included in an additional note, when mentioning the Hywind Scotland pilot park.

2. *page 3, line 50. Beware: Advanced-spar may be protected by copyright by JMU (Japan-Marine-United) as it is the name of their concept.*

   The authors could not find any information on the question if the term "advanced spar" is protected by copyright by JMU. However, it is ensured that the pure term "advanced spar" is only used in relation to the Fukushima Floating Offshore Wind Farm Demonstration Project FORWARD and otherwise the terms "advanced spar-type …" or "advanced geometry spar" are used throughout the paper.

3. *page 3, lines 56-59. Does this mean that the structural arrangement is not considered in the optimization process ? Rewording may be useful.*

   The original sentence "The focus of the optimization procedure lies on hydrodynamic and system-level analyses and no further limitations regarding a high detail structural design are added." is reformulated into "The focus of the optimization procedure lies on hydrodynamic and system-level analyses and not that stringent limitations on the structure and dimensions are required.".

4. *page 4, lines 104-107. It may be pointed that the ballasting operations of this unit proved complex: the hull accidentally listed more then 30degrees when it was brought to a deeper draught than the construction draught.*

   A sentence on this issue is added.

5. *page 6, lines 170-172. The ratio of structure mass to volume on the Hywind demonstrator is in excess of 0.17.*

   A note is added that the Hywind demonstrator is for safety reasons oversized and the given ratio of 0.13 is based on representative values from research designs and academic studies.

6. *page 13, lines 315-317. Load cases with transient loads (grid loss + gust) usually give rise to high accelerations and loads. It would be useful to clarify why they were not considered.*

   This aspect is addressed in the discussion chapter by additional remarks added at the end of the first main paragraph in Chapter 6.

7. *page 13, Table 3. Although sufficient for the demonstration of the optimisation method, using only one wave period may not be sufficient to capture the influence of the change of natural periods in the iterative optimisation process. A discussion / warning on this point should be added in the paper.*

   This aspect is addressed in the discussion chapter by additional remarks added at the end of the first main paragraph in Chapter 6.

8. *page 14, line 334. Simulation times as low as 600s do not allow to capture the low frequency dynamics of the floating wind turbines. It is understood that this is sufficient for the purpose of demonstration of the process, but this cannot be considered in the design process of a structure to be built.*

   This aspect is addressed in the discussion chapter by additional remarks added at the end of the first main paragraph in Chapter 6.

9. *page 27, lines 573-577. Or using plated partial bulkheads for loads transfer.*

   This aspect is included in an additional note.

---

## Referee Report (RR1)

Review – A fully integrated optimization framework for designing a complex geometry offshore wind turbine spar-type floating support structure

This work considers hull steel mass minimization for a spar floating wind turbine (FWT) using a genetic algorithm. The topic of FWT optimization is certainly of general interest, and the approach is interesting. There are, however, several important shortcomings in the present analysis. Some of these are discussed, but some additional analysis would greatly strengthen the presented results.

One of the main concerns relates to the estimation of the hydrodynamic loads on these generic hull forms. It is difficult to accept that the MacCamy-Fuchs formulation + Froude-Krylov forces in heave will give representative loads for these geometries with multiple horizontal surfaces. At the very least, the hydrodynamic characteristics of the optimized design should be studied in i.e. WAMIT or NEMOH, and a comparison of the performance should be given. This is discussed to some extent in section 6, but a re-analysis would provide much more information.

The approach for selecting the wall thickness in the present work may also be questioned. The steel mass per displaced volume is selected based on traditional spar designs, and yet applied to very different designs. At a minimum, hydrostatic pressure and the horizontal plates (top and bottom of the cylindrical sections) need to be considered. For the selected design, if I understand correctly, there are significant areas of the outer structure (which are subjected to hydrostatic and hydrodynamic pressures) which are simply accounted for by a very thin cap. This means that the optimizer will unrealistically reward designs with large diameter. In reality, such a design will require stiffeners and bulkheads (as well as expensive welding for the truss section which might replace the middle part of the column).

The mooring system assumptions are also confusing to me: are the fairlead locations maintained at z=-70m regardless of the draft of the design? This also has important consequences for the mean pitch motions.

Some additional information about the optimizer would also strengthen the present work. For example, how are the variables coded? What strategies are employed to introduce variation (mutation, immigration, others)? Could the performance be improved by "culling" the initial population so that (at a minimum) the geometric constraints which are cheap to compute are satisfied? It would be nice to distinguish between bounds and constraints in the optimization definition.

The introduction/text should be updated to account for the state of industry FWT farms (i.e. WindFloat Atlantic).

The paragraph beginning on line 35 is rather unwieldy and could be shortened – perhaps a table or other approach could be used to summarize the literature in a more efficient way? At a minimum, this paragraph should be separated into several shorter paragraphs.

I think it would make the reader's life easier if table and figures referred to physical variable names (for example $D_{BC_{up}}$) rather than optimization variable names $x_i$, which are more difficult to remember.

In general, the paper would also benefit from an effort to shorten and simplify the sentences.

---

## Editor Decision (ED1)

The paper still is lacking in terms of scientific content/quality for publication and requires additional work. As it is, the paper would be OK for a conference with the emphasis more on the chain of tools that they use rather than on the actual problem solved. The mooring system is not treated and the overall problem is quite simple with out of the box tools that are interconnected. There is a good amount of prior art in this space and it is unclear that this work really moves beyond the state of the art in a substantial way. For spar optimization, also including the controller in some cases, there is some work out there that is not fully addressed by the authors. Some examples:

- Hegseth JM, Bachynski EE. 2019. A semi-analytical frequency domain model for efficient design evaluation of spar floating wind turbines. Marine Structures
- Dou S, Pegalajar-Jurado A, Wang S, Bredmose H, Stolpe M. 2020. Optimization of floating wind turbine support structures using frequency-domain analysis and analytical gradients. Journal of Physics: Conference Series
- Souza CES, Hegseth JM, Bachynski EE. 2020. Frequency-dependent aerodynamic damping and inertia in linearized dynamic analysis of floating wind turbines. Journal of Physics: Conference Series
- Hegseth JM, Bachynski EE, Martins JR. 2020. Integrated design optimization of spar floating wind turbines. Marine Structures

Please consider extending the analysis complexity and/or demonstrating more clearly how this work extends substantially beyond the state-of-the-art.

Also, there are some typos – make sure to do another proofread before resubmitting.

More detailed notes include:
- Page 8, advanced – strong adjective for simple formulation
- Page 8, by addressing… - you could get sub optimal designs, or less optimized since the optimal solutions can not be achieved with a guarantee (GA is used)
- Page 9, advanced - why advanced? seems a simple sizing problem with few variables, mooring is excluded
- Page 9, as, however, this distribution… - repetition
- Page 11, optimized advanced spar-type… - see prior comments
- Page 11, fully-coupled complex floating offshore wind turbine system – what do you mean by complex?
- Page 12, x1, the diameter of BCup – lot of page space – why?
- Page 12, advanced – see prior comments
- Page 13, complex optimization problem with seven design variables and 25 constraints – again, not complex
- Page 18, nsgaII… - these are typically used for multi-objective optimization FYI
- Page 21, development of the design variables… the problem converges very quickly, 20 iterations is very small
- Page 22, advanced – see prior comments, need to justify this better
- Page 26, development of the objective function throughout the iterative optimization process, again, converged in 20 iterations.... what are the convergence criteria then for the opt analysis?
- Page 26, zooming into the objective - for a genetic alg it is quite a simple problem... only 7 variables and mostly linear constraints (not of system response). the challenge here is the system response evaluation done externally with a Modellica model
- Page 30, these – the results?
- Page 33, advanced – remove the use of the word advanced or quality further why it s so, the optimization itself is not demonstrably advanced compared to the state of the art

---

## Author Response (AR2)

The authors would like to thank the reviewers for the constructive recommendations and comments which will help improve the current and future work. In the following, the authors would like to respond to the reviewers' comments. The addressed comments are included in italic font.

**Answers to reviewer 1:**

1. T*he paper still is lacking in terms of scientific content/quality for publication and requires additional work. As it is, the paper would be OK for a conference with the emphasis more on the chain of tools that they use rather than on the actual problem solved. The mooring system is not treated and the overall problem is quite simple with out of the box tools that are interconnected. There is a good amount of prior art in this space and it is unclear that this work really moves beyond the state of the art in a substantial way. For spar optimization, also including the controller in some cases, there is some work out there that is not fully addressed by the authors. Some examples:*
   - *Hegseth JM, Bachynski EE. 2019. A semi-analytical frequency domain model for efficient design evaluation of spar floating wind turbines. Marine Structures*
   - *Dou S, Pegalajar-Jurado A, Wang S, Bredmose H, Stolpe M. 2020. Optimization of floating wind turbine support structures using frequency-domain analysis and analytical gradients. Journal of Physics: Conference Series*
   - *Souza CES, Hegseth JM, Bachynski EE. 2020. Frequency-dependent aerodynamic damping and inertia in linearized dynamic analysis of floating wind turbines. Journal of Physics: Conference Series*
   - *Hegseth JM, Bachynski EE, Martins JR. 2020. Integrated design optimization of spar floating wind turbines. Marine Structures*

   *Please consider extending the analysis complexity and/or demonstrating more clearly how this work extends substantially beyond the state-of-the-art.*

   > The paper goes beyond the common approaches for spar optimization, not by focusing on including more aspects but rather by considering alternative – more innovative and novel – design solutions. This novel content of the paper and novel approach, which is different to existing studies and work in the literature, is emphasized in more detail by adding in the introduction section, as well as in the abstract, that novel structural realization approaches are considered for the resulting optimized geometries and also alternative ballast materials are taken into account. The final sentence in the abstract is, furthermore, reformulated into:
   >> Thus, the presented design optimization example emphasizes the advantage of following a freer optimization formulation and allowing for novel structural approaches, by which means innovative floater designs, optimized with respect to the global system performance, can be obtained.
   >
   > Furthermore, the following paragraph is added at the end of Section 2.1, which demonstrates clearly how this work extends current and existing research and goes beyond the common approaches:
   >> Within this study, however, the definition of an advanced spar-type floater is further extended and goes beyond the main objectives to reduce the draft of the floater and the cost of the overall system. Thus, additionally, alternative materials are investigated, which are from an economic point of view comparative to currently used materials, however, positively influence the final

floater design due to their different material properties and characteristics. Furthermore, the term advanced spar-type floater - used in this study - not only addresses the floating structure itself, but also includes the consideration of novel structural approaches which might be more promising than the common approach of welding cylindrical and tapered sections together and allow a widening of the design space for such innovative and advanced floater designs. The specific steps taken for addressing the definition of an advanced spar-type floater in a broader sense are described in detail in Sect. 2.4.

With respect to the exemplary additional literature, proposed by the reviewer, these are included in the paper as follows: (Hegseth and Bachynski, 2019) does not focus on an optimization approach, however, addresses the aspect of a reduced draft spar-buoy floater and, hence, is referenced in the third paragraph of the introduction. (Hegseth et al., 2020) is referenced in several parts of the paper: 1) in the introduction section, presenting an optimization approach, which aims for a reduced draft, focuses on several aspects and components, uses gradient-based methods, and includes limits on the maximum allowable taper angle for conventional manufacturing approaches; 2) in Section 5.4, as 15° are considered as maximum inclination angle for a parked spar-type floating wind turbine system in extreme environmental conditions; and 3) in the discussion chapter, addressing the considered limits on the maximum allowable taper angle based on conventional manufacturing approaches. (Dou et al., 2020) is added now as well in the introduction section, as this addresses the optimization of a spar-buoy.

2. *Also, there are some typos – make sure to do another proofread before resubmitting.*

   Throughout the paper some typos are corrected and the sentences are simplified, shortened, or split up into several separate sentences to improve the readability of the paper.

3. *More detailed notes include:*
   - *Page 8, advanced – strong adjective for simple formulation*
   - *Page 9, advanced - why advanced? seems a simple sizing problem with few variables, mooring is excluded*
   - *Page 11, optimized advanced spar-type… - see prior comments*
   - *Page 12, advanced – see prior comments*
   - *Page 22, advanced – see prior comments, need to justify this better*
   - *Page 33, advanced – remove the use of the word advanced or quality further why it s so, the optimization itself is not demonstrably advanced compared to the state of the art*

   The following paragraph is added at the end of Section 2.1, which demonstrates clearly why the strong adjective "advanced" is used in this paper and what specifically is meant and comprised in this term:

   Within this study, however, the definition of an advanced spar-type floater is further extended and goes beyond the main objectives to reduce the draft of the floater and the cost of the overall system. Thus, additionally, alternative materials are investigated, which are from an economic point of view comparative to currently used materials, however, positively influence the final

floater design due to their different material properties and characteristics. Furthermore, the term advanced spar-type floater - used in this study - not only addresses the floating structure itself, but also includes the consideration of novel structural approaches which might be more promising than the common approach of welding cylindrical and tapered sections together and allow a widening of the design space for such innovative and advanced floater designs. The specific steps taken for addressing the definition of an advanced spar-type floater in a broader sense are described in detail in Sect. 2.4.

4. *More detailed notes include:*
   - *Page 8, by addressing… - you could get sub optimal designs, or less optimized since the optimal solutions can not be achieved with a guarantee (GA is used)*

   The beneficial properties of using NSGAII are outlined in Section 4.3.2. There, it is also added that evolutionary algorithms are highly suited to find the global optimum of a defined optimization problem for such a complex engineering system, as a floating wind turbine is – based on the following added reference:
   > Mishra, S.; Sahoo, S.; Das, M. Genetic Algorithm: An Efficient Tool for Global Optimization. Adv. Comput. Sci. Technol. 2017, 10, 2201–2211.

5. *More detailed notes include:*
   - *Page 9, as, however, this distribution… - repetition*

   The repetition is removed and the two consecutive sentences are rephrased as follows:
   > The resulting allowable total height of the BC has to be distributed to the three partitions; however, no restrictions prevail and also the option of utilizing not all three BC parts is possible. Thus, the minimum allowable value for the height of each of the BC parts is machine epsilon ($10^{-15}$ m) - as a zero value is unfeasible from a modeling point of view.

6. *More detailed notes include:*
   - *Page 11, fully-coupled complex floating offshore wind turbine system – what do you mean by complex?*

   The word complex is removed there, as the term "fully-coupled" already implies the complexity of such a system and the coupled motions and system responses.

7. *More detailed notes include:*
   - *Page 12, x1, the diameter of BCup – lot of page space – why?*

   The space was just due to the enumeration of the single design variables. As more information on the design variables (including a formal expression, the allowable value ranges, as well as the corresponding constraints) are added, the list is transformed into

a table (now Table 2. Definition of the seven design variables.) and the page space no longer exists.

8. *More detailed notes include:*
   - *Page 18, nsgaII… - these are typically used for multi-objective optimization FYI*

   Yes, this aspect is addressed in Section 4.3.2, where it is also stated that such a genetic algorithm can deal with both formulations of an optimization problem: single-objective and multi-objective. Thus, NSGAII can also be used for this single-objective optimization problem and at the same time it is taken benefit from the high suitability of NSGAII for such a floating wind turbine system optimization problem, as well as from the good performance and capability of parallelization in a highly efficient manner – as stated in the first paragraph of Section 4.3.2.

9. *More detailed notes include:*
   - *Page 30, these – the results?*

   The sentence is rephrased as follows:
   > In addition to the results presented, analyzed, and discussed in Sect. 5, more details on these results are addressed in the following and further aspects are discussed.

10. *More detailed notes include:*
    - *Page 13, complex optimization problem with seven design variables and 25 constraints – again, not complex*
    - *Page 21, development of the design variables… the problem converges very quickly, 20 iterations is very small*
    - *Page 26, development of the objective function throughout the iterative optimization process, again, converged in 20 iterations.… what are the convergence criteria then for the opt analysis?*
    - *Page 26, zooming into the objective - for a genetic alg it is quite a simple problem… only 7 variables and mostly linear constraints (not of system response). the challenge here is the system response evaluation done externally with a Modellica model*

    The complexity of the presented optimization problem is considered in comparison to the first-stage design optimization application example (Section 4.3.2). In this section, it is also stated that "the convergence is checked separately when post-processing the simulation results". Thus, the optimum solution is taken based on the individual, which exhibits the lowest value for the structure material volume and at the same time complies with all constraints (Section 5.3.). Even if, as the reviewer states, the objective function has already converged significantly after around 20 iterations, there is still a large spread in some of the design variables (and also the objective function), including as well several individuals per generation that do not comply with all constraints.

**Answers to reviewer 2:**

1. *One of the main concerns relates to the estimation of the hydrodynamic loads on these generic hull forms. It is difficult to accept that the MacCamy-Fuchs formulation + Froude-Krylov forces in heave will give representative loads for these geometries with multiple horizontal surfaces. At the very least, the hydrodynamic characteristics of the optimized design should be studied in i.e. WAMIT or NEMOH, and a comparison of the performance should be given. This is discussed to some extent in section 6, but a re-analysis would provide much more information.*

    The authors agree with the reviewer that a more detailed hydrodynamic analysis would be required for such a completely different shape. Based on the hull shape, the obtained optimum floater design is right now lying between the common floater designs and ship structures. Thus, the authors furthermore believe that a separate sensitivity study would be required to elaborate the relevance and degree of necessity of advanced panel-based tools to be used for the design development of such innovative and novel floater designs. Such a detailed sensitivity study, utilizing and comparing different tools, like the currently used ones, the suggested tools WAMIT or NEMOH, or even CFD, however, goes beyond the scope of this paper and would rather be the scope of a separate stand-alone subsequent research study and paper.

2. *The approach for selecting the wall thickness in the present work may also be questioned. The steel mass per displaced volume is selected based on traditional spar designs, and yet applied to very different designs. At a minimum, hydrostatic pressure and the horizontal plates (top and bottom of the cylindrical sections) need to be considered. For the selected design, if I understand correctly, there are significant areas of the outer structure (which are subjected to hydrostatic and hydrodynamic pressures) which are simply accounted for by a very thin cap. This means that the optimizer will unrealistically reward designs with large diameter. In reality, such a design will require stiffeners and bulkheads (as well as expensive welding for the truss section which might replace the middle part of the column).*

    The authors understand the reviewer's concern and agree with the comment raised. As stated in the paper (among others in Section 5.3 and Chapter 6), the obtained optimum design "would not directly be technically feasible, both from a manufacturing point of view and with respect to structural integrity" and "plated partial bulkheads for load transfer" would be required to be added. The main purpose of this rather freer optimization approach is to allow for a widened design space and to enable the detection of alternative design solutions, which are better performing from a global system performance and cost point of view compared to the common designs that are more stringently restricted, as only conventional manufacturing approaches (welding cylindrical and tapered sections together) are allowed. The final design is not yet to be taken as the final realistic solution. However, this will serve as basis for discussions with manufacturers, what options exist and which ways can be taken to realize structures of such or similar shapes. With these inputs and information, a second optimization round – taking new constraints for such alternative manufacturing solutions into account – has to be performed subsequently to find the final optimum, but still novel floater design solution.

3. *The mooring system assumptions are also confusing to me: are the fairlead locations maintained at z=-70m regardless of the draft of the design? This also has important consequences for the mean pitch motions.*

> The resulting mooring system properties from the original system design are taken and used, still accounting for the motion of the floater (last paragraph of Section 2.3). As discussed in the second paragraph in Chapter 6, a subsequent optimization of the mooring system properties and layout design can further improve the performance of the floating system, including the pitch motion – corresponding to the system inclination.

4. *Some additional information about the optimizer would also strengthen the present work. For example, how are the variables coded? What strategies are employed to introduce variation (mutation, immigration, others)? Could the performance be improved by "culling" the initial population so that (at a minimum) the geometric constraints which are cheap to compute are satisfied? It would be nice to distinguish between bounds and constraints in the optimization definition.*

- The coding of the variables is straight-forward, as the design variables are variables which exist in the model. The same is valid for the other variables addressed in the constraints and the objective function. These are determined within the numerical model and defined as outputs of the simulation, so that these can directly be evaluated by the optimizer, following the equations presented in the paper.
- With respect to the details for the strategies for representing the evolution, the following item is added to the bullet point list in Section 4.3.2:
  > The individuals are randomly generated. When evaluating the objective function and constraints, the dominant individuals - each selected based on a comparison of two individuals - form the basis for the next generation, which is created without using any variator. These are the default generator, selector, and variator settings of NSGAII in Platypus.

  Furthermore, the additional information that "the tournament selector for evaluating the dominance is used" is added in Section 4.3.3.
- With respect to the suggested improvement of the performance, the authors have to state that the geometric constraints are already satisfied, based on the followed approach. This is due to the fact, that the optimizer selects the new individuals just based on the allowable value ranges, specified in the corresponding constraints. This is described in Section 4.3.3. This is also again stated in Section 5.1: "The first 14 constraints for the allowable value ranges of the design variables are excluded, as they are not constraints that are evaluated after the simulation but are taken into account ahead of the simulations when the optimizer selects the design variables for the new individuals and, hence, are never violated."
- The definition of the optimization problem, given at the beginning of Section 3, follows the commonly used formal description, using the design variable vector, the objective functions, as well as the constraints, where the bounds are also included. To better distinguish between bounds and constraints, the enumeration of the

design variables in Section 3.1 is changed into a table (now Table 2. Definition of the seven design variables), in which more details, such as the allowable value ranges – hence, the bounds – as well as the corresponding constraints, are provided. Furthermore, in the table presenting the constraints (now Table 3), it becomes clear from the description that $g_1$ to $g_{14}$ are constraints specifying the bounds for the design variables and, additionally, $g_{21}$ and $g_{22}$ are the constraining bounds for the ballast density.

5. *The introduction/text should be updated to account for the state of industry FWT farms (i.e. WindFloat Atlantic).*

    The introduction is updated, considering the recent technology and industrial steps that had happened since the last submission of the revised paper. Thus, WindFloat Atlantic is included, the overall number of floating foundation concepts is updated, and also the expected date for the TetraSpar demonstrator installation is updated.

6. *The paragraph beginning on line 35 is rather unwieldy and could be shortened – perhaps a table or other approach could be used to summarize the literature in a more efficient way? At a minimum, this paragraph should be separated into several shorter paragraphs.*

    This paragraph had been extended based on the request from other reviewers for a more in-depth and detailed literature review. The authors can fully understand the reviewer's concern. Thus, the paragraph is broken down into single shorter paragraphs and structured as follows:
    - the relevance for enhancing the common spar-buoy design from a more general point of view;
    - approaches for advanced spar-buoy floater designs by modifying the floater itself;
    - approaches for advanced spar-buoy floater designs adding and modifying other components (and not the floating structure itself);
    - common design optimization approaches for enhancing the common spar-buoy design;
    - going beyond the common approaches by considering novel structural realization approaches and alternative ballast materials – the approach presented in this paper).

7. *I think it would make the reader's life easier if table and figures referred to physical variable names (for example DBCup) rather than optimization variable names xi, which are more difficult to remember.*

    To make the transfer between the design variables $x_i$ and corresponding physical variable names easier, the list of the design variables in Section 3.1 is changed into a table (now Table 2. Definition of the seven design variables), in which also the formal expression is added. Thus, the reader does not need to search for any description in the

text but can just look up the corresponding physical variable name and description in the table.

8. *In general, the paper would also benefit from an effort to shorten and simplify the sentences.*

Throughout the paper some typos are corrected and the sentences are simplified, shortened, or split up into several separate sentences to improve the readability of the paper.

---

## Editor Decision (ED2)

WES 2020-93

Overview

Generally, this paper is improved, but it still needs some further improvements before publication. Overall, what is the point of the work? It is a huge effort to bring dynamic analysis inside of a conceptual design optimization process. Would an approach using static analysis be able to capture the effects on constraints and allow for similar exploration of the design space without the heavy computation? The key for showing this paper is value is that including the in situ dynamic analysis adds realism that a static analysis would lack and thus drives the designs differently.

The results of section 4 essentially show limitations of the problem formulation (i.e. it is missing critical design constraints). Recasting these oversights as a good thing (i.e. that they suggest truss spar solutions) is spurious. The paper is doing spar not truss-spar optimization – the conclusion that the optimization hints towards superiority of truss-spar configurations is not justifiable. Instead of this, there should be some discussion on what using the computationally costly dynamic analysis has provided. It's still not clear to me and this is core to the paper.

Additional high-level notes:
- It is not clear until section 3.3 what you are actually doing in the study. Saying modular / fully integrated / multi-fidelity optimization etc is abstract and obfuscates what you are doing. Please in plain language both in abstract and introduction say what you are doing succinctly – something like: "In this study, we perform conceptual design optimization for a FOWT spar with *in situ* aero-hydro-elastic simulations."
- Please have a native English speaker due a detailed edit if possible. There are still various parts of the paper that are hard to read / difficult to understand because of the English grammar problems. There are many typos and grammatical errors. Apologies that I do not have time to go through all of them.
- The paper is still very long – some of the text is run-on discussion. An editor could also help with clarity of message and brevity. Having short, concise and clear text and messaging is always better.

More detailed notes by section follow below.

Abstract

- The first sentences talk about a multi-fidelity approach from conceptual to detailed design but only the results for the conceptual design are discussed in the abstract (and paper as far as II can tell)

Introduction
- Small floating wind farms – you mean first generation commercial floating wind farms? Small is ambiguous
- Semi-subs are also deployed. Arguing spar is most mature is not necessarily true and not so important to the paper
- Figure 1: I don't think it is good to show low resolution graphics from other papers… instead, create your own graphic that illustrates the basic features of the most common topologies for spars
- Sentence in lines 53 to 61 is a long run-on sentence and is hard to read. Please break it down and make sure it is clear
- "aimed and obtained", "aimed to be obtained"… rephrase, this is not clear

- What is in and out of scope should be brought up at the end of the introduction– i.e. the lack of inclusion of the mooring system. The scope of the paper is still unclear by the time I get to the end of the introduction – it is confused by the promise in the abstract for the multi-fidelity framework.

Forming the basis for innovative floater configurations
- A table of properties – initial and final – could help in section 2.2
- Section 2.3 is unnecessarily verbose – can you use tables and simplify the text? Section 2.3 and should be combined into section 3, and then streamlined and shortened as section 2.3 has everything to do with the problem set-up

Optimization problem
- See notes on 2.3 – interweave the 2.3 points into section 3

Fully modular and automated design optimization
- The paragraph on DLCs can be truncated. Since you are using the IEC, just state that you are using DLCs x,y,z from standard a,b,c. The audience is very familiar with the design standards and you don't need to explain them.
- In section 3.3.2 there is far too much detail on the optimizer. For a simple structure as a spar, there are many studies that show gradient-based algorithms work well. I do not think you should try to argue that the system is so complex you must use GA or that GA is actually the best choice. Rather, say that for this study you simply used the Gas as these were available in the framework. Future work may look at alternative algorithms and further improvement of the optimization implementation. GA algoirithms are also well known – there is again far too much detail explaining the algorithm. Simply point to a reference on the algorithm is fine. What should be detailed is your specific parameterization of the algorithm (i.e. population size, etc) which you do. A short paragraph would be enough.
- Optimization algorithm section is a bit of a strange label. The optimization algorithm is the GA. What you describe in 3.3.3 is your optimization workflow.
- Section 3.3.3 is the first section where it is really clear what you are actually doing in the optimization. Consider bringing this forward to the introduction and abstract but state it plainly, something like this:
  - "In this study, we use in situ aero-hydro-elastic simulations to support conceptual design optimization for a FOWT spar."
- Can you say anything in terms of how often simulations fail? I imagine it could be a lot.
- I do not think you need equations 5, 6 or 7 at all… this is trivial information about how optimization works in general

Results
- 31 days!!! That is a heck of a lot of time… and you are not converged at all? It is hard to tell because in figure 4 you start with the initial population which has a much larger spread in performance compared to the subsequent populations.
- For figure 4 it would be nice to see the lables in terms of the formal expression rather than design variables so the reader gets a better feel for what they are without having to refer back
- I wouldn't refer to the final design as the "selected optimum" – you know its not optimal. Say instead the best performing individual from the final population
- Description of what plot colors mean should be in the caption, not in the text.
- Same thing for figure 5, label the plots with the formal expression and not the constraint function variables
- Same thing in table 6, use formal expression for DVs, constraints and objective function

- It is not clear if "these individuals" as in line 401 on page 4.2 are from the first generation or final generation. As they obey the constraints (as far as I can see), they are from the final population? Make sure that it is explicit. How did you select these individuals? Is it random? And is their ordering random? Their objective function performance is wildly different- if it were ordered from lowest to highest
- In figure 6, you can see that there are a lot of really weird designs. It is quite easy to implement basic manufacturability constraints for example which would help avoid getting such weird designs. Having such weird results undermines to a degree the overall credibility of the work – this is dangerous as an industry person looking at these results would likely be very dismissive of the work.
- The comparison of the design results to figure 7 also is strange. Truss type spars are different than what you are modelling. There are simulation tools out there to design such things, but saying that your spar framework can help identify structures like truss that could be sought is a big leap… its also a big leap to draw a link between the example designs and the two truss spars from the literature. I don't think you can do so based on the results you have.
- "one and only objective function" is strange wording… just say objective function. Delete the entire first sentence of section 4.3 – this is how basic optimization works and does not need explanation.
- There is a lot of redundancy in section 4.1 through 4.3 – you can much more quickly get to the point around the final best performing individual and eliminate making overreaching statements about the capability.
- The basic issues with the final design related to lack of constraints related to manufacturability and/or structural analysis should have been corrected before publication as they are easy to implement and appear in various prior studies (for example tower, jacket and monopile optimization from NREL, Stuttgart, and elsewhere). I do not think rebranding it as hinting towards truss spar solutions is justifiable. If the paper were to do truss spar optimization it should do it and would need to reformulate the model to account for the truss elements – in other words, it would be an entirely different study. **Instead of focusing on the truss, please expand on what using the dynamic analysis in situ has added in value compared to using static or quasi-static analysis**.

Discussion
- Recommend shortening substantially the discussion section. They are far too detailed. Every paragraph in the section including the bulleted paragraphs can be shortened likely by half. Be clear and concise.
- The lines from 605 on overstate things. It would be better to say that future work will incorporate constraints on stress, buckling and manufacturability to ensure that the designs are realistic but also allowing for the exploration of a wide range of novel concepts.
- I don't think you need to speak to LCOE optimization unless you are going to speak to the elements across LCOE that the spar design impacts… i.e. where the couplings will be that require a more holistic approach

Conclusion
- The last sentence is still an overreach… what can you conclude from this work? In one to two sentences

---

## Author Response (AR3)

**Answers to the reviewer:**

The authors would like to thank the reviewer for the constructive recommendations and comments which will help improve the current and future work. In the following, the authors would like to respond to the reviewer's comments. The addressed comments are included in italic font.

1. *The manuscript contains the statement "advanced spar-type". In my opinion, there is nothing advanced with either the spar or the optimization problem and the authors should moderate these statements. The proposed approach can perhaps be used for conceptual design studies, but both the optimization problem and the simulations lack capabilities to perform more detailed design. For example, manufacturing considerations and cost, structural requirements, and mooring system design are all excluded. The authors are encouraged to extend the work in at least one of these directions. It may be that the argument that less restrictive feasible sets can provide unexpected designs, but it is clear from the figures that several of the presented designs have structural integrity issues (this is also acknowledged in the manuscript).*

   The phrase advanced floater is avoided and rephrased into advancements taken for achieving a conceptual innovative floater design (e.g. alternative structural realization approaches, alternative ballast materials, …). It is emphasized that the applied approach only aims to obtain a conceptual design, however, that a detailed design optimization can be performed with the same framework due to the fully modular optimization problem setup and the multi-fidelity numerical modelling and optimization environment. These aspects are more clearly stated and emphasized throughout the paper. Especially, most of the abstract is rewritten, so that the focus is directly put into the right light so that the messaging on the intention of the paper and applied approach, as well as on the value of the work and contrast to other approaches is clearer positioned and emphasized. Furthermore, Figures on the referenced "advanced" floater configurations are added to improve the better understanding of the intended innovative character and approach.

2. *Unsuccessful simulations are encountered frequently according to the manuscript. This is an important topic within simulation-based optimization and appropriate actions must be taken. When unsuccessful simulations are flagged, they are dealt with in the implementation by considering the design under study as infeasible. The frequency of unsuccessful simulations, the type, and severity of failures are not reported. This could lead to a very conservative approach, with possibilities of disregarding good designs. The manuscript should be extended to report on the unsuccessful simulations and the authors are encouraged to investigate the reasons, types, and severities for the failures.*

   More details and information are provided on the unsuccessful simulations. Thus, it is made clear that these failing designs were not good designs which are disregarded but unstable designs with negative metacentric height. These explanations are provided in Section 4.3.3, when defining the undesired parameter values for failing simulations, and given as well when presenting and discussing the results in Section 5.1.

3.  *The description of the optimization problem is unnecessarily verbose and spread out over several sections. It is for example not necessary to describe bounds on design variables in detail like in Table 3. The same holds for the other linear constraints. The focus should be on the constraints that require simulations. For the presented optimal designs, it is notable that almost none of the constraints are active, which suggests that the problem without simulations should also be solved and the designs compared. The authors should also investigate if the solver has actually found an optimal design.*

    Duplications in the derivation of the assessment criteria (Section 2) and the resulting definition of the optimization problem (Section 3) are removed. The general description of the optimization problem, however, is left complete in the paper, as this was as well asked by other reviewers and is the basis for defining the optimization problem. Based on Figure 5, it becomes clear that most of the constraints are often violated or close to the limit. This is also emphasized by the fact that the number of individuals that comply with all constraints (marked with darker colored markers in Figures 4, 5, and 8) is significantly lower than all simulated individuals. In Section 5.3 in the paragraph from line 484 to 497 it is elaborated in detail on the convergence of the optimization and the found optimum design solution.

4.  *Several of the bounds in the optimization problem are physically unrealistic and additionally likely to cause numerical issues in the simulations. The problems should be re-solved with realistic values.*

    The specified allowable value ranges for the design variables do not cause numerical issues in the simulations. The only reason for failing simulations are design solutions which are unstable due to a negative metacentric height, as explained in the answer to the second comment of the reviewer. In Section 2.2, detailed information on the chosen allowable value ranges and corresponding reasons and argumentation are provided. Furthermore, it is elaborated on alternative "physical" realizations of solutions and values, which are initially deemed as unphysical. This is part of the intention of the paper and presented approach, namely that this freer optimization formulation can shed the light on innovative floater configurations, which require alternative structural realization approaches, of which some are already utilized in highly innovative floater concepts, such as the TetraSpar or Hexafloat. Thus, more details and examples are added and provided in Section 5.2.

5.  *The reported optimization (wall-clock) time is more than 31 days. This is clearly far too much for conceptual design studies. The authors should make an effort to reduce this or at the very least explain the reasons.*

    More details on the duration of the optimization are added:
    - In Section 5.3 the following sentences are added:
      If, in addition to the maximum number of simulations, a reasonable convergence tolerance had been specified as supplementary stop criterion, the optimization algorithm would not have required all 10,000 simulations and would have stopped

much earlier. However, due to the strong similarity of the last design solutions, no significant differences in the results would have been perceived.

- The following separate paragraph is added in Section 6:
First of all, the duration of the optimization simulations needs to be addressed. If an additional stop criterion based on a realistic convergence tolerance has been specified, not the full 10,000 simulations would have to be simulated as the convergence tolerance would have been reached maybe already after around 40 generations. Thus, the conceptual design study would have required maybe just less than a quarter of the actually spent time. However, even around 181 hours - which is more than a week - is still too long for just a conceptual design study. The reason behind the currently quite long time required does not lie in the multi-fidelity framework and fully modular optimization problem setup, but rather in the developmental stage of the numerical model for a floating wind turbine system. Thus, a 800 s load case simulation with a floating wind turbine in irregular sea state and with turbulent wind conditions takes about four and a half hours, which is about 20 times as much as the time to be simulated. This is a known issue and part of the current development work at Fraunhofer IWES. While for bottom-fixed wind turbine systems real-time capability of the numerical models based on MoWiT has already been achieved (Feja and Huhn, 2019), the optimization of the code for floating systems is still at an early stage of development. However, based on the experience with the bottom-fixed numerical wind turbine system models, it is expected to achieve as well real-time capability for floating numerical wind turbine system models based on MoWiT after some further advancements of the code. At that time, the full simulation of the specified optimization problem will just require about one and a half days. This would then be very promising both for conceptual design studies, as well as detailed design optimization tasks, which - due to the fully modular and multi-fidelity approach applied - can be realized with the same numerical modeling and optimization environment.

6. *The found mass is compared to an existing design and it is reasoned that significant material reductions can be achieved. It is however not reported if the two designs are subject to the same design requirements and have both been assessed in the same way. The authors should address these topics in detail before stating conclusions.*

The authors are aware of the fact that the OC3 phase IV spar-buoy floating wind turbine system was defined as a reference design for code-to-code verifications and code-to-experiments validation and, hence, was not necessarily optimized. The statement about the mass, provided in Section 5.3, is mainly and simply to clarify, that alternative configurations can be found by applying the presented procedure. The general aspect that the OC3 phase IV design was not designed with the same design requirements and was not optimized, as it mainly dealt as reference floating wind turbine system design, is added in the paper:

- In Appendix A, referring to Section 2.1, the following sentence is added:
This OC3 phase IV spar-buoy floating wind turbine system was defined as a reference design for code-to-code verifications and code-to-experiments validation and, hence, was not necessarily yet optimized.

- In the fourth paragraph of Section 5.3 the sentence with the statement on the mass reduction is extended as follows:
  This design solution yields a reduction of the structure material volume of more than 31% compared to the original (modified) reference spar-type floating platform, for which it must be noted that it neither has been designed with the same design requirements, nor has it yet been optimized.

7. *The manuscript can with advantage be drastically shortened, sharpened, and re-organized. Several sections contain information already presented.*

   In general, the paper has been tightened and a bit restructured. Thus, the introduction now only contains the main information relevant to the content of the paper (the list of spar floaters previously included in the introduction is fully removed). A paper roadmap is included, incorporating the information previously provided in both introduction and the separate chapter on advanced spar-type floater and their characteristics. Duplications in the derivation of the assessment criteria and the resulting definition of the optimization problem are removed. Furthermore, bullet point lists are removed and, if required, tables are used instead. The details for the (already well-known) OC3 phase IV floating wind turbine system are put to the appendix.

---

## Author Response (AR4)

**Answers to the reviewer:**

The authors would like to thank the reviewer for the constructive recommendations and comments which will help improve the current and future work. In the following, the authors would like to respond to the reviewer's comments. The addressed comments are included in italic font.

1. *There is a mismatch between the many simplifying assumptions and the claims of a "fully integrated framework for designing complex geometry". The fact that the turbine, mooring system, and structural analysis are all excluded and that some of the hydrodynamic constants are not updated with design changes means that the work oversells itself. Hence the unfamiliar designs in the results seem even more surprising given the claims earlier in the paper.*

   - The focus of the paper is further clarified by emphasizing that the applied approach targets a conceptual but more innovative design by incorporating advanced features, while the fully modular and multi-fidelity character of the numerical modelling and optimization environment allow for further extension to address as well holistic design optimization task of higher fidelity. Thus, the following paragraph in the introduction is modified and extended:

        Thus, this paper aims to demonstrate that through a freer optimization formulation more potential solutions for an advanced spar-type floater design with a higher degree of innovation can be captured. The conceptual design study and optimization approach, applied in this work, focus on hydrodynamic and system-level analyses. Due to the conceptual character of this study, which precisely targets to explore novel design spaces, not that stringent limitations on the structure and dimensions are yet required. The optimization approach followed in this paper bases on an initial design optimization example by Leimeister et al. (2020b), which, however, is quite simple and does not include any aspects and goals for going beyond and advancing the common spar-buoy floater design but only focuses on optimizing the global system performance. While global system performance criteria still have to be fulfilled but are only incorporated as constraints, the main objective of this study is cost reduction – expressed in terms of the material used – and the optimization problem is specified in such a way that advancements, which go beyond just obtaining a reduced draft, can be achieved. Hence, by allowing design variables out of a wider range of values, contemplating different ballast materials, and considering novel structural realization approaches for the resulting optimized geometries, new alternatives of potential and innovative floater design solutions are opened up. All these requirements regarding design variables and optimization criteria are – together with specific environmental conditions and the fully coupled aero-hydro-servo-elastic dynamic characteristics of a FOWT system – incorporated into a fully modular optimization framework. Its current capability is sufficient for this conceptual design study; however, due to its close interlinking with the – as well – fully modular and multi-fidelity numerical modeling environment, the framework can easily be extended for serving more holistic FOWT system design optimization problems of higher fidelity, including a subsequent detailed design development.

- With respect to the reviewer's comment on the hydrodynamic constants, which are not all updated, the authors want to refer to the details provided in the discussion and want to emphasize that "the horizontal added mass coefficient, as well as the total inertia force, are already determined in dependency of the actual structural diameter and wave number" for each column element separately. Only "the horizontal drag coefficient is currently not altered from the original value of 0.6. This is a valid assumption for large diameters already at low flow velocities" and, hence, a valid assumption as well for the optimized design solution obtained within this work.
- Indeed, the designs in the result are quite unfamiliar, however, the can indicate potential design solutions, which might be more cost-efficient, however, require innovative structural realization approaches and alternative features. More details on ways to realize these "unfamiliar" designs, including the presentation of real examples of such innovative design solutions, which might deal as idea and impulse provider for alternative structural realization approaches, have been added and included in the introduction and Section 4.2 (which was previously Section 5.2) in the revised paper version from 5th June 2021.

2. *With that idea in mind, the Introduction does a poor job of setting up the paper. Too much text is devoted to a market survey and not enough to explaining the premise of the paper. For instance, the Introduction makes statements that assume spars are difficult to handle (line 47) and that "advanced" spars can address those shortcomings. However, these concepts are not explained until he beginning of Section 2. Also, given the prior publication by the authors (https://www.sciencedirect.com/science/article/pii/S0029801820302286) I would also recommend including the explanation of how this article goes beyond it. I would state explicitly what the contribution is to the literature in this paper.*

Some comments by the reviewer do not refer to the latest version of the paper (submitted on 5th June 2021) but still address the previous version from 22nd March 2021. In the revised paper from 5th June 2021, the following changes compared to the previous version had already been made, which address the reviewer's comments in the first four sentences:
- The market survey, mentioning relevant milestones of the floating wind technology development, has been completely removed.
- The comment regarding line 47 and "the statements that assume spars are difficult to handle and that 'advanced' spars can address those shortcomings" refers to the previous version of the paper, submitted on 22nd March 2021.
- In the paper version from 22nd March 2021, there has been indeed the definition of such "advanced" spars just in Section 2; however, in the revised paper from 5th June 2021, this part from Section 2 has been incorporated into the Introduction to clarify already at that point what "advanced" solutions might comprise.

To further address the reviewer's first comment, as well as the remaining to regarding the other publication by the authors and the contribution, the following additional changes are done: The focus and contribution of the paper are further clarified and the prior publication by the authors is addressed already in the introduction, specifying how this paper goes beyond the previous work and what the contribution of this paper to research and literature is. Thus, the following paragraph in the introduction is modified and extended:

Thus, this paper aims to demonstrate that through a freer optimization formulation more potential solutions for an advanced spar-type floater design with a higher degree of innovation can be captured. The conceptual design study and optimization approach, applied in this work, focus on hydrodynamic and system-level analyses. Due to the conceptual character of this study, which precisely targets to explore novel design spaces, not that stringent limitations on the structure and dimensions are yet required. The optimization approach followed in this paper bases on an initial design optimization example by Leimeister et al. (2020b), which, however, is quite simple and does not include any aspects and goals for going beyond and advancing the common spar-buoy floater design but only focuses on optimizing the global system performance. While global system performance criteria still have to be fulfilled but are only incorporated as constraints, the main objective of this study is cost reduction – expressed in terms of the material used – and the optimization problem is specified in such a way that advancements, which go beyond just obtaining a reduced draft, can be achieved. Hence, by allowing design variables out of a wider range of values, contemplating different ballast materials, and considering novel structural realization approaches for the resulting optimized geometries, new alternatives of potential and innovative floater design solutions are opened up. All these requirements regarding design variables and optimization criteria are – together with specific environmental conditions and the fully coupled aero-hydro-servo-elastic dynamic characteristics of a FOWT system – incorporated into a fully modular optimization framework. Its current capability is sufficient for this conceptual design study; however, due to its close interlinking with the – as well – fully modular and multi-fidelity numerical modeling environment, the framework can easily be extended for serving more holistic FOWT system design optimization problems of higher fidelity, including a subsequent detailed design development.

3. *The article is still unnecessarily long and many statements are repeated section by section. For instance, both sections 2.3, 2.4, 3.\*, explain the design variables and constraints. These should be condensed and combined. For Sections 2 & 4, if the approach or methods are the same as in prior publications, it might be easiest to provide a very terse description and cite the prior work. Finally, the paper could be much more succinct by reading the paper through and determining if each sentence/paragraph contributes to core narrative or not. For instance, some of the comments about the computational execution history (first paragraph in Section 5) are extraneous and the plots of the GA results (Fig 3,4,6) say very little to the reader. I could list many more examples. If professional editing is available, that could help quite a bit as well.*

Some comments by the reviewer do not refer to the latest version of the paper (submitted on 5[th] June 2021) but still address the previous version from 22[nd] March 2021. In the revised paper from 5[th] June 2021, for example, Section 2 only comprises subsections up to 2.3. Thus, there is no longer a Section 2.4., which, however, the reviewer refers to. As the first subsection of Section 2 was removed and shifted to the introduction in the revision from 5[th] June 2021, the authors transfer the reviewer's comments regarding Sections 2.3 and 2.4 to the Sections 2.2 and 2.3. Furthermore, the reviewer refers to Figs. 3, 4, and 6 as plots of the GA results. Due to the additional

figures included in the revised paper from 5[th] June 2021, the authors derive from the comments on the "plots of the GA results" that the reviewer refers to Figs. 4, 5, and 8. Based on this, the following changes are done:

- The specification of the optimization problem, which was originally covered separately in Section 3, is now directly incorporated in Section 2 and the design variables, objective function, and optimization constraints are directly derived from the specifications in Sections 2.2. and 2.3.
- Apart from the same basis floating wind turbine system, which is since the last revision from 5[th] June 2021 already only very shortly specified in Section 2.1 (with some more details on the parameters moved to the appendix), and the same three values for the constraints on the global system performance – as well only covered very briefly in the final paragraph of Section 2.3, all methods and approaches presented in Section 2 are new and have not yet been covered in any prior work.
- Section 4 (which is now Section 3) is shortened and details from prior work, which are not necessarily to be repeated and presented in this paper again, are removed to condense the text.
- The extraneous information and comments on the computational execution history at the beginning of Section 4 (which was previously Section 5) are removed. The plots of the GA results (Figs. 4, 5, and 8), however, are kept in the paper, as the support the reader in understanding the development process described textually in Section 4.1.
- Throughout the paper, modifications are made and not elementary parts are removed to condense the entire paper.

---

## Author Response (AR5)

**Answers to the comments**

Wednesday 8th December 2021

Dear Editor, dear Katherine

On behalf of the authors of this paper, I want to thank you for your additional review and detailed comments. We have addressed your comments in the revision of the paper. These are highlighted in the submitted PDF and detailed below, referring to your comments included in italic font.

1. *Overview*
   *Generally, this paper is improved, but it still needs some further improvements before publication. Overall, what is the point of the work? It is a huge effort to bring dynamic analysis inside of a conceptual design optimization process. Would an approach using static analysis be able to capture the effects on constraints and allow for similar exploration of the design space without the heavy computation? The key for showing this paper is value is that including the in situ dynamic analysis adds realism that a static analysis would lack and thus drives the designs differently.*

   > Since "static analysis" in terms of using ultimate loads only does not comply with all the current literature, where at least a frequency approach is needed to capture the oscillatory load, "static analysis" is understood rather as a "quasi-static" frequency approach. With this understanding the following additions are made, pointing out the benefits of applying such a time-domain analysis instead of a frequency-domain one, e.g., inclusion of transient loads and non-linear loads.
   > In the first sentences of the abstract the following changes are made:
   > > Spar-type platforms for floating offshore wind turbines are considered suitable for commercial wind farm deployment. To reduce the hurdles of such floating systems becoming competitive, **in-situ aero-hydro-servo-elastic simulations are applied to support conceptual design optimization by including transient and non-linear loads. For reasons of flexibility, the utilized optimization framework and problem are modularly structured so that the setup can be applied to both an initial** conceptual design study for bringing innovative floater configurations to light and a subsequent optimization for obtaining detailed designs.
   > In the second last paragraph of the introduction the following changes are made:
   > > Thus, this paper aims to demonstrate that through a freer optimization formulation **with in-situ aero-hydro-servo-elastic simulations,** more potential solutions for an advanced spar-type floater design with a higher degree of innovation can be captured**, while already including transient and non-linear loads in the analysis**.
   > In the first part of the conclusion the following changes are made:
   > > In this paper, an automated optimization approach is applied to a spar-type FOWT system to develop a conceptual innovative floating platform design, which is optimized with respect to the change in hydrodynamics and their impact on the main system performance, while structural, manufacturability, or other constraints are not considered, whereas other advancements are

facilitated. This approach, following a freer optimization formulation **with in-situ aero-hydro-servo-elastic simulations to include transient and non-linear loads already in the system analyses**, is taken in order to be able to explore novel design spaces **that** can be better from a hydrodynamic point of view and show potential for more cost-efficient design solutions, but may require novel structural approaches.

2. *Overview*
*The results of section 4 essentially show limitations of the problem formulation (i.e. it is missing critical design constraints). Recasting these oversights as a good thing (i.e. that they suggest truss spar solutions) is spurious. The paper is doing spar not truss-spar optimization – the conclusion that the optimization hints towards superiority of truss-spar configurations is not justifiable. Instead of this, there should be some discussion on what using the computationally costly dynamic analysis has provided. It's still not clear to me and this is core to the paper.*

The key objective of the approach and corresponding results is that the design space that is set to be investigated is not limited to (conventional) spar-type floaters, but is let open enough to investigate alternative solutions, still within the modeling capabilities of the utilized framework.

The authors agree that if a conventional spar floater design is to be obtained and the traditional manufacturing processes of welding cylindrical sections together are to be followed, the basic manufacturability constraints should be directly implemented in the optimization problem. However, because this application example is specifically aimed at expanding the design space and not being limited to the conventional floater design, such basic manufacturability constraints are purposefully left out of the optimization problem. This approach lets the optimizer explore novel configurations that are not necessarily covered by conventional floater manufacturing techniques. The resulting designs could be of similar shapes as existing solutions or very different as well. Thus, when setting up the optimization problem, it was also not intended to obtain a truss type spar, but it was left very open in which direction the design would develop from a hydrodynamic point of view.

Due to the manner in which the single cylindrical elements are implemented in the numerical model and connected to each other, the resulting shapes may really look weird. However, the results and potential design solutions found are not expected to be realized as is since it is just a conceptual design optimization study. The shapes should indicate what would be best from a hydrodynamic and system-level performance point of view. Based on these results, it is to think about alternative manufacturing solutions and discuss with manufacturers the corresponding manufacturing constraints, which are then to be integrated into the optimization problem for the subsequent detailed design optimization. The inclusion of references to recent innovative design solutions, such as the TetraSpar or Hexafloat, will assist industry professionals in recognizing the potential.

As an aside:
Michael Borg from Stiesdal was one of my examiners when defending my doctoral thesis. He was especially very interested in this approach and very positive about the results, especially that the optimization yielded a design solution that exhibits similarity to Stiesdal's TetraSpar.

Furthermore, in the first revision round of the paper, Thomas Choisnet from Ideol was one of the reviewers (he commented in the PDF, for which reason his name was identifiable). Based on his comments, it became clear that he also saw the potential for the industry. He even suggested, to add to the potential alternative structural realization methods, that "plated partial bulkheads for loads transfer" could be used, which was hence included in the first revision.

3. *Additional high-level notes*
   *It is not clear until section 3.3 what you are actually doing in the study. Saying modular / fully integrated / multi-fidelity optimization etc is abstract and obfuscates what you are doing. Please in plain language both in abstract and introduction say what you are doing succinctly – something like: "In this study, we perform conceptual design optimization for a FOWT spar with in situ aero-hydro-elastic simulations."*

   > The first sentences of the abstract are restructured and rephrased so that it now becomes clear that only the conceptual design optimization is applied and presented in this paper. Thus, the first sentences now read as follows:
   > > Spar-type platforms for floating offshore wind turbines are considered suitable for commercial wind farm deployment. To reduce the hurdles of such floating systems becoming competitive, **in-situ aero-hydro-servo-elastic simulations are applied to support conceptual design optimization by including transient and non-linear loads. For reasons of flexibility, the utilized optimization framework and problem are modularly structured so that the setup can be applied to both an initial** conceptual design study for bringing innovative floater configurations to light and a subsequent optimization for obtaining detailed designs.
   > Furthermore, it is added at a later point in the abstract that it is only the "conceptual" (and not the detailed) design optimization presented.
   > > The approach for generating an initial but very innovative **conceptual** floater design comprises…
   > In the second last paragraph of the introduction, in which the objective and content of the paper are described, some clarifications are added, e.g.:
   > > Thus, this paper aims to demonstrate that through a freer optimization formulation **with in-situ aero-hydro-servo-elastic simulations,** more potential solutions for an advanced spar-type floater design with a higher degree of innovation can be captured**, while already including transient and non-linear loads in the analysis**.

4. *Additional high-level notes*
   *Please have a native English speaker due a detailed edit if possible. There are still various parts of the paper that are hard to read / difficult to understand because of the English grammar problems. There are many typos and grammatical errors. Apologies that I do not have time to go through all of them.*

The entire paper is checked for grammar and spelling, and corrections are made throughout the paper. Furthermore, long sentences that are difficult to read are broken down into several shorter sentences.

5. *Additional high-level notes*
*The paper is still very long – some of the text is run-on discussion. An editor could also help with clarity of message and brevity. Having short, concise and clear text and messaging is always better.*

The entire paper is revised focusing on shortening and removing duplicate information or information and discussion that stray too far from the topic of the paper. In particular, significant reductions are done in section 2 and 3 (combining the optimization problem definition with the assessment criteria), 4.1 on the DLCs and 4.3 on both the optimizer and the optimization workflow, 5 on the results (removing duplicate information that is already contained in figures or was mentioned before), and 6 on the discussion.

6. *Abstract*
*The first sentences talk about a multi-fidelity approach from conceptual to detailed design but only the results for the conceptual design are discussed in the abstract (and paper as far as II can tell)*

The first sentences of the abstract are restructured and rephrased so that it now becomes clear that only the conceptual design optimization is applied and presented in this paper. Thus, the first sentences now read as follows:
> Spar-type platforms for floating offshore wind turbines are considered suitable for commercial wind farm deployment. To reduce the hurdles of such floating systems becoming competitive, **in-situ aero-hydro-servo-elastic simulations are applied to support conceptual design optimization by including transient and non-linear loads. For reasons of flexibility, the utilized optimization framework and problem are modularly structured so that the setup can be applied to both an initial** conceptual design study for bringing innovative floater configurations to light and a subsequent optimization for obtaining detailed designs.

Furthermore, it is added at a later point that it is only the "conceptual" (and not the detailed) design optimization presented.
> The approach for generating an initial but very innovative **conceptual** floater design comprises…

7. *Introduction*
*- Small floating wind farms – you mean first generation commercial floating wind farms? Small is ambiguous*
The phrase "small floating wind farms" is replaced by "pilot floating wind farms".

*- Semi-subs are also deployed. Arguing spar is most mature is not necessarily true and not so important to the paper*

> This part is removed, and the first sentences of the second paragraph are shortened.

*- Figure 1: I don't think it is good to show low resolution graphics from other papers… instead, create your own graphic that illustrates the basic features of the most common topologies for spars*

> Since the textual description is sufficient for describing briefly some common "advanced" spar-type floater topologies and as the focus should lie on the geometric basis used for the conceptual design optimization presented in Figure 2 (which is now Figure 1), the illustrations (which did not even cover the most common topologies addressed in the paragraph) are removed.

*- Sentence in lines 53 to 61 is a long run-on sentence and is hard to read. Please break it down and make sure it is clear*

> The sentence is split into three shorter sentences that are easier to read.

*- "aimed and obtained", "aimed to be obtained"… rephrase, this is not clear*

> This phrase is rephrased as "aimed at".

*- What is in and out of scope should be brought up at the end of the introduction– i.e. the lack of inclusion of the mooring system. The scope of the paper is still unclear by the time I get to the end of the introduction – it is confused by the promise in the abstract for the multi-fidelity framework.*

> In the second last paragraph of the introduction, in which the objective and content of the paper are described, some clarifications are added, e.g.:
>
> > Thus, this paper aims to demonstrate that through a freer optimization formulation **with in-situ aero-hydro-servo-elastic simulations,** more potential solutions for an advanced spar-type floater design with a higher degree of innovation can be captured**, while already including transient and non-linear loads in the analysis**.
>
> Furthermore, it is directly highlighted what is in and out of scope, as added in the subsequent sentence:
>
> > The conceptual design study and optimization approach, applied in this work, focus on hydrodynamic and system-level analyses **but do not yet include an optimization of the mooring system**.

8. *Forming the basis for innovative floater configurations*
   *- A table of properties – initial and final – could help in section 2.2*

   > A table is added, comparing the original FOWT system parameter values with those of the initially adjusted system that is used as an initial design for the optimization.

   *- Section 2.3 is unnecessarily verbose – can you use tables and simplify the text? Section 2.3 and should be combined into section 3, and then streamlined and shortened as section 2.3 has everything to do with the problem set-up*

   > The optimization problem is moved out of section 3 and merged with the content of section 2.3 into a separate section. This is required since the performance criteria are

introduced in these parts but already need to be known for the preprocessing analysis on selecting the most critical DLC. Thus, the separate section is put ahead of the section on the fully modular and automated design optimization. In this separate section, the single elements of the optimization problem are directly derived from the assessment criteria.

Furthermore, the content of section 2.3 is shortened since the table that summarizes the constraints derived from the assessment criteria is already included.

9. *Optimization problem*
   *See notes on 2.3 – interweave the 2.3 points into section 3*

   The optimization problem is moved out of section 3 and merged with the content of section 2.3 into a separate section. This is required since the performance criteria are introduced in these parts but already need to be known for the preprocessing analysis on selecting the most critical DLC. Thus, the separate section is put ahead of the section on the fully modular and automated design optimization. In this separate section, the single elements of the optimization problem are directly derived from the assessment criteria.

   Furthermore, the specification of the design variables in section 2.2 is as well moved to this optimization problem section.

10. *Fully modular and automated design optimization*
    *- The paragraph on DLCs can be truncated. Since you are using the IEC, just state that you are using DLCs x,y,z from standard a,b,c. The audience is very familiar with the design standards and you don't need to explain them.*

    The entire first part on existing standards is removed, and the paragraph on DLCs is shortened.

    *- In section 3.3.2 there is far too much detail on the optimizer. For a simple structure as a spar, there are many studies that show gradient-based algorithms work well. I do not think you should try to argue that the system is so complex you must use GA or that GA is actually the best choice. Rather, say that for this study you simply used the Gas as these were available in the framework. Future work may look at alternative algorithms and further improvement of the optimization implementation. GA algoirithms are also well known – there is again far too much detail explaining the algorithm. Simply point to a reference on the algorithm is fine. What should be detailed is your specific parameterization of the algorithm (i.e. population size, etc) which you do. A short paragraph would be enough.*

    The section on the optimizer is significantly shortened and reduced to one paragraph, focusing mainly on the specific parameterization utilized for the study.

    *- Optimization algorithm section is a bit of a strange label. The optimization algorithm is the GA. What you describe in 3.3.3 is your optimization workflow.*

    The term "optimization algorithm" is replaced by "optimization workflow" if the GA is not meant.

*- Section 3.3.3 is the first section where it is really clear what you are actually doing in the optimization. Consider bringing this forward to the introduction and abstract but state it plainly, something like this: "In this study, we use in situ aero-hydro-elastic simulations to support conceptual design optimization for a FOWT spar."*

> This information is added in the abstract and introduction as detailed in comments 3 and 6.

*- Can you say anything in terms of how often simulations fail? I imagine it could be a lot.*

> Failing simulations due to negative metacentric height and, hence, poor performance of instable designs occur mainly in the first few generations. Throughout the optimization, there are only around three individuals per generation that still fail to complete the entire simulation length. The number of failing systems becomes clear as well in the result plots for the constraints on the performance criteria, since there are only a few distinguishable results plotted in the first few generations.
> This information is now added in the results sections, when presenting the development of the constraints:
>
> > The fact that for **the performance constraints** $g_{15}$ to $g_{17}$**,** only a few distinguishable individuals are plotted in the first generations is caused by the large number of unstable design solutions that are selected by the optimizer in the first trials. Due to the unsuccessful simulations, the performance variables are set to undesired values, as explained in Sect. 4.3.2, and, hence, **they** are all the same for all failing systems. This is **also** visible throughout the generations, as there is a line at the specified undesired value formed by the individuals that do not complete the simulations successfully**, which, however, are only a few per generation (two to three in the higher generations)**.

*- I do not think you need equations 5, 6 or 7 at all... this is trivial information about how optimization works in general*

> Based on your suggestion, and since the approach is already stated sufficiently clearly in the test, the equations are removed.

11. *Results*
    *- 31 days!!! That is a heck of a lot of time... and you are not converged at all? It is hard to tell because in figure 4 you start with the initial population which has a much larger spread in performance compared to the subsequent populations.*

    > The "non-convergence" stated previously in the paper was formulated in such a way that it was misleading. Since no convergence tolerance is specified as a constraint, all 10,000 individuals needed to be simulated. Considering the convergence, the conceptual design optimization would have taken about a week. This long duration is just due to the not yet improved numerical code of the FOWT system. Right now, the focus is on improving the numerical model and code. After this, it is expected that the conceptual design optimization will take less than two days.
    > Some clarification is given in section 5.3:
    >
    > > The fact that this optimum solution is just found in the last generation states that **the optimizer still tries to improve the result for the objective function since no convergence tolerance has been specified as a stop criterion and the 10,000 simulations have to be completed.**

Furthermore, this issue is discussed in detail in the first paragraph of section 6:

First of all, the duration of the optimization simulations needs to be dealt with. If an additional stop criterion based on a realistic convergence tolerance had been specified, **only a fraction of** the 10,000 simulations would have **had** to be simulated as the convergence tolerance would have been reached already after around 40 generations. Thus, the conceptual design study would have required just less than a quarter of the actual spent time. However, even around 181 hours---which is more than a week---is still too long for just a conceptual design study**, which should take no more than two days**. The reason behind the currently quite long time required does not lie in the multi-fidelity framework and fully modular optimization problem setup, but rather in the developmental stage of the numerical model for a FOWT system[*]. While for bottom-fixed wind turbine systems**,** real-time capability of the numerical models based on MoWiT has already been achieved (Feja and Huhn, 2019), the optimization of the code for floating systems is still at an early stage of development. **When this is achieved**, the full simulation of the specified optimization problem will **only** require about one and a half days.

[*]**An** 800~s load case simulation with a FOWT in **an** irregular sea state and with turbulent wind conditions takes about four and a half hours, which is about 20 times as much as the time to be simulated.

*- For figure 4 it would be nice to see the lables in terms of the formal expression rather than design variables so the reader gets a better feel for what they are without having to refer back*

The formal expressions of the design variables are used in figure 4.

Furthermore, the legend is removed, and the description of the different colors used in the plots is added in the caption.

*- I wouldn't refer to the final design as the "selected optimum" – you know its not optimal. Say instead the best performing individual from the final population*

The term "best-performing individual" is used instead of the term "selected optimum".

*- Description of what plot colors mean should be in the caption, not in the text.*

This is done for figures 3, 4, 5, 7, and 8 (updated numbers).

*- Same thing for figure 5, label the plots with the formal expression and not the constraint function variables*

The formal expressions of the constraints are used in figure 5.

Furthermore, the legend is removed, and the description of the different colors used in the plots is added in the caption.

*- Same thing in table 6, use formal expression for DVs, constraints and objective function*

The formal expressions of the design variables, constraints, and objective function are used in table 6.

This is done in table 8 as well.

*- It is not clear if "these individuals" as in line 401 on page 4.2 are from the first generation or final generation. As they obey the constraints (as far as I can see), they are from the final population? Make sure that it is explicit. How did you select these individuals? Is it random? And*

*is their ordering random? Their objective function performance is wildly different- if it were ordered from lowest to highest*

It is clarified that only the individuals that comply with all optimization constraints are further assessed in this section. However, it is also clarified that the focus does not yet lie on the objective function result but rather on showing the wide diversity of potential innovative floater geometries.

The geometric design variables of these individuals **that meet all constraints** are **presented** in Fig. 5. From these individuals **that** comply with all constraints, seven examples **out of different generations** are selected to demonstrate the diversity of potential innovative floater geometries, **not yet focusing on their performance with respect to the objective function**.

The individuals that comply with all constraints originate from almost all generations (apart from the first), which becomes clear in the development plots (the dark recolored crosses), as well as in table 7 by the figures corresponding to the selected exemplary geometries.

*- In figure 6, you can see that there are a lot of really weird designs. It is quite easy to implement basic manufacturability constraints for example which would help avoid getting such weird designs. Having such weird results undermines to a degree the overall credibility of the work – this is dangerous as an industry person looking at these results would likely be very dismissive of the work.*

Some detailed answers to this are provided in the answer to comment 2. For further clarification, the objective of the approach is added here as well:

Looking at the floater geometries presented in Fig. 5, it becomes clear that not all of these shapes can be realized with conventional manufacturing solutions, where cylindrical sections are welded together. It has to be emphasized that these results are solely based on the hydrodynamic and system-level analyses, as specified within the optimization problem, as well as on the advancements taken into account in Sect. 3**,** which clearly intend the utilization of alternative and innovative structural realization approaches **and let the optimizer explore novel configurations that are not necessarily covered by conventional floater manufacturing techniques**.

Furthermore, the formal expressions of the design variables are used in figure 6. Additionally, the legend is removed, and the description of the different colors used in the plots is added in the caption.

*- The comparison of the design results to figure 7 also is strange. Truss type spars are different than what you are modelling. There are simulation tools out there to design such things, but saying that your spar framework can help identify structures like truss that could be sought is a big leap… its also a big leap to draw a link between the example designs and the two truss spars from the literature. I don't think you can do so based on the results you have.*

The approach does not aim at truss type spars or any other specific geometries, but rather left the design space and potential geometry open. The reference to the designs presented in figure (now) 6 shall help industry professionals recognize the potential of this "freed" approach. The authors refer to the detailed answer given to comment 2.

*- "one and only objective function" is strange wording… just say objective function. Delete the entire first sentence of section 4.3 – this is how basic optimization works and does not need explanation.*

> Yes, considering the background of the readership, the first paragraph is not required and hence deleted.

*- There is a lot of redundancy in section 4.1 through 4.3 – you can much more quickly get to the point around the final best performing individual and eliminate making overreaching statements about the capability.*

> Section 4 is shortened and condensed. Unnecessary information, e.g., on the different colors used in the plots, is removed.

*- The basic issues with the final design related to lack of constraints related to manufacturability and/or structural analysis should have been corrected before publication as they are easy to implement and appear in various prior studies (for example tower, jacket and monopile optimization from NREL, Stuttgart, and elsewhere). I do not think rebranding it as hinting towards truss spar solutions is justifiable. If the paper were to do truss spar optimization it should do it and would need to reformulate the model to account for the truss elements – in other words, it would be an entirely different study. **Instead of focusing on the truss, please expand on what using the dynamic analysis in situ has added in value compared to using static or quasi-static analysis.***

> Some detailed answers to this are provided in the answer to comment 2.

12. *Discussion*
*- Recommend shortening substantially the discussion section. They are far too detailed. Every paragraph in the section including the bulleted paragraphs can be shortened likely by half. Be clear and concise.*

> All parts of the discussion section are condensed. Duplicate information or discussions that stray too far from the topic of the paper are removed.

*- The lines from 605 on overstate things. It would be better to say that future work will incorporate constraints on stress, buckling and manufacturability to ensure that the designs are realistic but also allowing for the exploration of a wide range of novel concepts.*

> The last sentence of this paragraph has been removed, and it is more focused on the future work on incorporating structural integrity checks and constraints on stress and buckling.

*- I don't think you need to speak to LCOE optimization unless you are going to speak to the elements across LCOE that the spar design impacts… i.e. where the couplings will be that require a more holistic approach*

> Since the final paragraph on addressing LCoE optimization is too far from the topic of the paper, it is removed entirely from the discussion.

13. *Conclusion*
*The last sentence is still an overreach… what can you conclude from this work? In one to two sentences*

The last sentence of the conclusion on the multi-fidelity framework is deleted and two final concluding sentences are added, focusing more on the overall objective of the paper and the results:

> Thus, the presented approach of expanding the design space and purposefully leaving out basic manufacturability constraints in the conceptual design study lets the optimizer explore novel configurations that are not necessarily covered by conventional floater manufacturing techniques. The results of the presented conceptual design optimization exhibit similarities to recent innovative design solutions, such as Stiesdal's TetraSpar and Saipem's Hexafloat, which emphasizes the potential for the industry.

---

## Author Response (AR6)

**A fully integrated optimization framework for designing a complex geometry offshore wind turbine spar-type floating support structure**

Mareike Leimeister[1,2], Maurizio Collu[1], and Athanasios Kolios[1]

[1]Naval Architecture, Ocean and Marine Engineering, University of Strathclyde, 100 Montrose Street, Glasgow G4 0LZ, United Kingdom
[2]Division System Technology, Fraunhofer IWES, Institute for Wind Energy Systems, Am Luneort 100, 27572 Bremerhaven, Germany

**Correspondence:** Mareike Leimeister (mareike.leimeister@iwes.fraunhofer.de)

Thursday 6th January 2022

Dear Editor, dear Katherine

On behalf of the authors of this paper, I want to thank you for handling our manuscript. We are pleased that it was now accepted for publication.

In the author's certification on the file upload step, it is written that we "certify that the content of all files uploaded in this form is exactly the same as the version of my manuscript accepted by the Associate Editor". Since the rights owner of the Hexafloat picture has granted the permission to use the figure, however, has requested to use it without the detailed picture included on the anchor, I have adjusted this accordingly. Thus, I just wanted to notify you that this is the only change made compared to the version that got accepted. For completing the file upload, I tick the box on the author's certification.

Best Regards,

Mareike Leimeister